# CLIENT SELECTION IN FEDERATED LEARNING: CONVERGENCE ANALYSIS AND POWER-OF-CHOICE SELECTION STRATEGIES

## ABSTRACT

Federated learning is a distributed optimization paradigm that enables a large number of resource-limited client nodes to cooperatively train a model without data sharing. Several works have analyzed the convergence of federated learning by accounting of data heterogeneity, communication and computation limitations, and partial client participation. However, they assume unbiased client participation, where clients are selected at random or in proportion of their data sizes. In this paper, we present the first convergence analysis of federated optimization for biased client selection strategies, and quantify how the selection skew affects convergence speed. We reveal that biasing client selection towards clients with higher local loss achieves faster error convergence. Using this insight, we propose POWER-OF-CHOICE, a communication- and computation-efficient client selection framework that can flexibly span the trade-off between convergence speed and solution bias. We also propose an extension of POWER-OF-CHOICE that is able to maintain convergence speed improvement while diminishing the selection skew. Our experiments demonstrate that POWER-OF-CHOICE strategies can converge up to $3\times$ faster and give $10\%$ higher test accuracy than the baseline random selection.

## 1 INTRODUCTION

Until recently, machine learning models were largely trained in the data center setting (Dean et al., 2012) using powerful computing nodes, fast inter-node communication links, and large centrally available training datasets. The future of machine learning lies in moving both data collection as well as model training to the edge. The emerging paradigm of federated learning (McMahan et al., 2017; Kairouz et al., 2019; Bonawitz et al., 2019) considers a large number of resource-constrained mobile devices that collect training data from their environment. Due to limited communication capabilities and privacy concerns, these data cannot be directly sent over to the cloud. Instead, the nodes locally perform a few iterations of training using local-update stochastic gradient descent (SGD) (Yu et al., 2018; Stich, 2018; Wang & Joshi, 2018; 2019), and only send model updates periodically to the aggregating cloud server. Besides communication limitations, the key scalability challenge faced by the federated learning framework is that the client nodes can have highly heterogeneous local datasets and computation speeds. The effect of data heterogeneity on the convergence of local-update SGD is analyzed in several recent works (Reddi et al., 2020; Haddadpour & Mahdavi, 2019; Khaled et al., 2020; Stich & Karimireddy, 2019; Woodworth et al., 2020; Koloskova et al., 2020; Huo et al., 2020; Zhang et al., 2020; Pathak & Wainwright, 2020; Malinovsky et al., 2020; Sahu et al., 2019) and methods to overcome the adverse effects of data and computational heterogeneity are proposed in (Sahu et al., 2019; Wang et al., 2020; Karimireddy et al., 2019), among others.

**Partial Client Participation.** Most of the recent works described above assume full client participation, that is, all nodes participate in every training round. In practice, only a small fraction of client nodes participate in each training round, which can exacerbate the adverse effects of data heterogeneity. While some existing convergence guarantees for full client participation and methods to tackle heterogeneity can be generalized to partial client participation (Li et al., 2020), these generalizations are limited to *unbiased* client participation, where each client's contribution to the expected global objective optimized in each round is proportional to its dataset size. In Ruan et al. (2020), the authors analyze the convergence with flexible device participation, where devices can freely join or leave the

training process or send incomplete updates to the server. However, adaptive client selection that is cognizant of the training progress at each client has not been understood yet.

It is important to analyze and understand biased client selection strategies since they can sharply accelerate error convergence, and hence boost communication efficiency in heterogeneous environments by preferentially selecting clients with higher local loss values, as we show in our paper. This idea has been explored in recent empirical studies (Goetz et al., 2019; Laguel et al., 2020; Ribero & Vikalo, 2020). Nishio & Yonetani (2019) proposed grouping clients based on hardware and wireless resources in order to save communication resources. Goetz et al. (2019) (which we include as a benchmark in our experiments) proposed client selection with local loss, and Ribero & Vikalo (2020) proposed utilizing the progression of clients' weights. But these schemes are limited to empirical demonstration without a rigorous analysis of how selection skew affects convergence speed.

Another relevant line of work (Jiang et al., 2019; Katharopoulos & Fleuret, 2018; Shah et al., 2020; Salehi et al., 2018) employs biased selection or importance sampling of data to speed-up convergence of classic centralized SGD – they propose preferentially selecting samples with highest loss or highest gradient norm to perform the next SGD iteration. In contrast, Shah et al. (2020) proposes biased selection of lower loss samples to improve robustness to outliers. Generalizing such strategies to the federated learning setting is a non-trivial and open problem because of the large-scale distributed and heterogeneous nature of the training data.

**Our Contributions.** In this paper, we present the first (to the best of our knowledge) convergence analysis of federated learning with biased client selection that is cognizant of the training progress at each client. We discover that biasing the client selection towards clients with higher local losses increases the rate of convergence compared to unbiased client selection. Using this insight, we propose the POWER-OF-CHOICE client selection strategy and show by extensive experiments that POWER-OF-CHOICE yields up to $3\times$ faster convergence with $10\%$ higher test performance than the standard federated averaging with random selection. POWER-OF-CHOICE is designed to incur minimal communication and computation overhead, enhancing resource efficiency in federated learning. In fact, we show that even with $3\times$ less clients participating in each round as compared to random selection, POWER-OF-CHOICE gives $2\times$ faster convergence and $5\%$ higher test accuracy.

## 2 PROBLEM FORMULATION

Consider a cross-device federated learning setup with total $K$ clients, where client $k$ has a local dataset $\mathcal{B}_k$ consisting $|\mathcal{B}_k| = D_k$ data samples. The clients are connected via a central aggregating server, and seek to collectively find the model parameter $\mathbf{w}$ that minimizes the empirical risk:

$$F(\mathbf{w}) = \frac{1}{\sum_{k=1}^{K} D_k} \sum_{k=1}^{K} \sum_{\xi \in \mathcal{B}_k} f(\mathbf{w}, \xi) = \sum_{k=1}^{K} p_k F_k(\mathbf{w}) \tag{1}$$

where $f(\mathbf{w}, \xi)$ is the composite loss function for sample $\xi$ and parameter vector $\mathbf{w}$. The term $p_k = D_k / \sum_{k=1}^{K} D_k$ is the fraction of data at the $k$-th client, and $F_k(\mathbf{w}) = \frac{1}{|\mathcal{B}_k|} \sum_{\xi \in \mathcal{B}_k} f(\mathbf{w}, \xi)$ is the local objective function of client $k$. In federated learning, the vectors $\mathbf{w}^*$, and $\mathbf{w}_k^*$ for $k = 1, \ldots, K$ that minimize $F(\mathbf{w})$ and $F_k(\mathbf{w})$ respectively can be very different from each other. We define $F^* = \min_{\mathbf{w}} F(\mathbf{w}) = F(\mathbf{w}^*)$ and $F_k^* = \min_{\mathbf{w}} F_k(\mathbf{w}) = F_k(\mathbf{w}_k^*)$.

**Federated Averaging with Partial Client Participation.** The most common algorithm to solve (1) is *federated averaging* (FedAvg) proposed in McMahan et al. (2017). The algorithm divides the training into communication rounds. At each round, to save communication cost at the central server, the global server only selects a fraction $C$ of $m = CK$ clients to participate in the training. Each selected/active client performs $\tau$ iterations of local SGD (Stich, 2018; Wang & Joshi, 2018; Yu et al., 2018) and sends its locally updated model back to the server. Then, the server updates the global model using the local models and broadcasts the global model to a new set of active clients.

Formally, we index the local SGD iterations with $t \geq 0$. The set of active clients at iteration $t$ is denoted by $\mathcal{S}^{(t)}$. Since active clients performs $\tau$ steps of local update, the active set $\mathcal{S}^{(t)}$ also remains constant for every $\tau$ iterations. That is, if $(t + 1) \bmod \tau = 0$, then $\mathcal{S}^{(t+1)} = \mathcal{S}^{(t+2)} = \cdots = \mathcal{S}^{(t+\tau)}$.

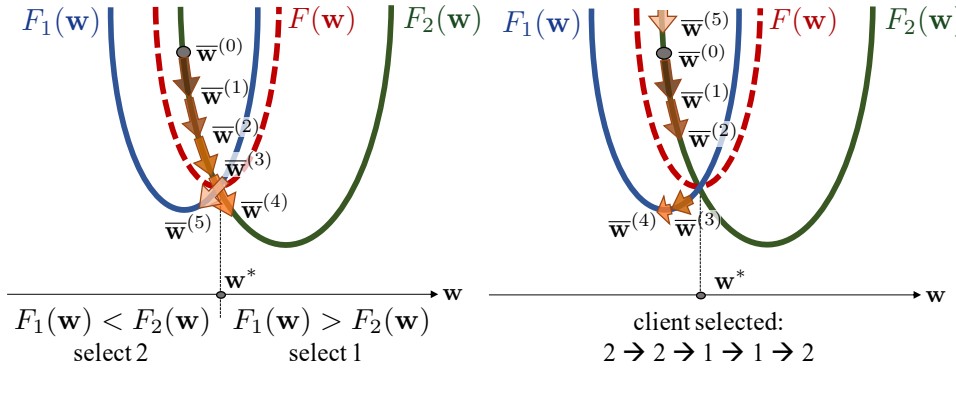

(a) Selecting Higher Loss Clients                    (b) Random Client Selection

Figure 1:  A toy example with $F_1(\mathbf{w})$, $F_2(\mathbf{w})$ as the local objective, and $F(\mathbf{w}) = (F_1(\mathbf{w}) + F_2(\mathbf{w}))/2$ as the global objective function with global minimum $\mathbf{w}^*$. At each round, only one client is selected to perform local updates. (a): Model updates for sampling clients with larger loss; (b): Model updates for sampling clients uniformly at random (we select client in the order of 2,2,1,1,2).

Accordingly, the update rule of FedAvg can be written as follows:

$$
\mathbf{w}_k^{(t+1)} =
\begin{cases}
\mathbf{w}_k^{(t)} - \eta_t g_k(\mathbf{w}_k^{(t)}, \xi_k^{(t)}) & \text{for } (t+1) \bmod \tau \neq 0 \\
\frac{1}{m} \sum_{j \in \mathcal{S}^{(t)}} \left( \mathbf{w}_j^{(t)} - \eta_t g_j(\mathbf{w}_j^{(t)}, \xi_j^{(t)}) \right) \triangleq \overline{\mathbf{w}}^{(t+1)} & \text{for } (t+1) \bmod \tau = 0
\end{cases}
\tag{2}
$$

where $\mathbf{w}_k^{(t+1)}$ denotes the local model parameters of client $k$ at iteration $t$, $\eta_t$ is the learning rate, and $g_k(\mathbf{w}_k^{(t)}, \xi_k^{(t)}) = \frac{1}{b} \sum_{\xi \in \xi_k^{(t)}} \nabla f(\mathbf{w}_k^{(t)}, \xi)$ is the stochastic gradient over mini-batch $\xi_k^{(t)}$ of size $b$ that is randomly sampled from client $k$'s local dataset $\mathcal{B}_k$. Moreover, $\overline{\mathbf{w}}^{(t+1)}$ denotes the global model at server. Although $\overline{\mathbf{w}}^{(t)}$ is only updated after every $\tau$ iterations, for the purpose of convergence analysis we consider a virtual sequence of $\overline{\mathbf{w}}^{(t)}$ that is updated at each iteration as follows

$$
\overline{\mathbf{w}}^{(t+1)} = \overline{\mathbf{w}}^{(t)} - \eta_t \overline{\mathbf{g}}^{(t)} = \overline{\mathbf{w}}^{(t)} - \eta_t \left( \frac{1}{m} \sum_{k \in \mathcal{S}^{(t)}} g_k(\mathbf{w}_k^{(t)}, \xi_k^{(t)}) \right)
\tag{3}
$$

with $\overline{\mathbf{g}}^{(t)} = \frac{1}{m} \sum_{k \in \mathcal{S}^{(t)}} g_k(\mathbf{w}_k^{(t)}, \xi_k^{(t)})$. Note that in (2) and (3) we do not weight the client models by their dataset fractions $p_k$ because $p_k$ is considered in the client selection scheme used to decide the set $\mathcal{S}^{(t)}$. Our convergence analysis can be generalized to when the global model is a weighted average instead of a simple average of client models, and we show in Appendix E that our convergence analysis also covers the sampling uniformly at random without replacement scheme proposed by Li et al. (2020). The set $\mathcal{S}^{(t)}$ can be sampled either with or without replacement. For sampling with replacement, we assume that multiple copies of the same client in the set $\mathcal{S}^{(t)}$ behave as different clients, that is, they perform local updates independently.

**Client Selection Strategy.** To guarantee FedAvg converges to the stationary points of the objective function (1), most current analysis frameworks (Li et al., 2020; Karimireddy et al., 2019; Wang et al., 2020) consider a strategy that selects the set $\mathcal{S}^{(t)}$ by sampling $m$ clients at random (with replacement) such that client $k$ is selected with probability $p_k$, the fraction of data at that client. This sampling scheme is *unbiased* since it ensures that in expectation, the update rule (3) is the same as full client participation. Hence, it enjoys the same convergence properties as local-update SGD methods (Stich, 2018; Wang & Joshi, 2018). We denote this unbiased random client selection strategy as $\pi_{\text{rand}}$.

In this paper, we consider a class of biased client selection strategies that is cognizant of the global training progress which (to the best of our knowledge) has not been worked on before. Note that for any aggregation scheme and sampling scheme with partial client participation, if the expectation over the sampling scheme for the update rule of the global model is equal to the case of the update rule for full client participation, we distinguish this as an *unbiased* client participation scheme. For example in Horváth & Richtárik (2020), even with a biased sampling scheme, with the normalizing aggregation

the update rule is unbiased. Henceforth, we state that our paper encompasses both biased and unbiased update rules. In the two-client example in Figure 1, we set $\mathcal{S}^{(t+1)} = \arg\max_{k \in [K]} F_k(\overline{\mathbf{w}}^{(t)})$, a single client with the highest local loss at the current global model. In this toy example, the selection strategy cannot guarantee the updates (3) equals to the full client participation case in expectation. Nevertheless, it gives faster convergence to the global minimum than the random one. Motivated by this observation, we define a client selection strategy $\pi$ as a function that maps the current global model $\mathbf{w}$ to a selected set of clients $\mathcal{S}(\pi, \mathbf{w})$.

## 3 CONVERGENCE ANALYSIS

In this section we analyze the convergence of federated averaging with partial device participation for *any* client selection strategy $\pi$ as defined above. This analysis reveals that biased client selection can give faster convergence, albeit at the risk of having a non-vanishing gap between the true optimum $\mathbf{w}^* = \arg\min F(\mathbf{w})$ and $\lim_{t\to\infty} \overline{\mathbf{w}}^{(t)}$. We use this insight in Section 4 to design client selection strategies that strike a balance between convergence speed and bias.

### 3.1 ASSUMPTIONS AND DEFINITIONS

First we introduce the assumptions and definitions utilized for our convergence analysis.

**Assumption 3.1.** $F_1, ..., F_k$ *are all* $L-$*smooth, i.e., for all* $\mathbf{v}$ *and* $\mathbf{w}$, $F_k(\mathbf{v}) \leq F_k(\mathbf{w}) + (\mathbf{v} - \mathbf{w})^T \nabla F_k(\mathbf{w}) + \frac{L}{2}\|\mathbf{v} - \mathbf{w}\|_2^2$.

**Assumption 3.2.** $F_1, ..., F_k$ *are all* $\mu-$*strongly convex, i.e., for all* $\mathbf{v}$ *and* $\mathbf{w}$, $F_k(\mathbf{v}) \geq F_k(\mathbf{w}) + (\mathbf{v} - \mathbf{w})^T \nabla F_k(\mathbf{w}) + \frac{\mu}{2}\|\mathbf{v} - \mathbf{w}\|_2^2$.

**Assumption 3.3.** *For the mini-batch* $\xi_k$ *uniformly sampled at random from* $\mathcal{B}_k$ *from user* $k$, *the resulting stochastic gradient is unbiased, that is,* $\mathbb{E}[g_k(\mathbf{w}_k, \xi_k)] = \nabla F_k(\mathbf{w}_k)$. *Also, the variance of stochastic gradients is bounded:* $\mathbb{E}\|g_k(\mathbf{w}_k, \xi_k) - \nabla F_k(\mathbf{w}_k)\|^2 \leq \sigma^2$ *for all* $k = 1, ..., K$.

**Assumption 3.4.** *The stochastic gradient's expected squared norm is uniformly bounded, i.e.,* $\mathbb{E}\|g_k(\mathbf{w}_k, \xi_k)\|^2 \leq G^2$ *for* $k = 1, ..., K$.

The above assumptions are common in related literature, see (Stich, 2018; Basu et al., 2019; Li et al., 2020; Ruan et al., 2020). Next, we introduce two metrics, the *local-global objective gap* and the *selection skew*, which feature prominently in the convergence analysis presented in Theorem 3.1.

**Definition 3.1 (Local-Global Objective Gap).** *For the global optimum* $\mathbf{w}^* = \arg\min_{\mathbf{w}} F(\mathbf{w})$ *and local optimum* $\mathbf{w}_k^* = \arg\min_{\mathbf{w}} F_k(\mathbf{w})$ *we define the local-global objective gap as*

$$\Gamma \triangleq F^* - \sum_{k=1}^{K} p_k F_k^* = \sum_{k=1}^{K} p_k(F_k(\mathbf{w}^*) - F_k(\mathbf{w}_k^*)) \geq 0. \tag{4}$$

Note that $\Gamma$ is an inherent property of the local and global objective functions, and it is independent of the client selection strategy. This definition was introduced in previous literature by Li et al. (2020). A larger $\Gamma$ implies higher data heterogeneity. If $\Gamma = 0$ then it implies that the local and global optimal values are consistent, and there is no solution bias due to the client selection strategy (see Theorem 3.1). Next, we define another metric called selection skew, which captures the effect of the client selection strategy on the local-global objective gap.

**Definition 3.2 (Selection Skew).** *For any* $k \in \mathcal{S}(\pi, \mathbf{w})$ *we define,*

$$\rho(\mathcal{S}(\pi, \mathbf{w}), \mathbf{w}') = \frac{\mathbb{E}_{\mathcal{S}(\pi, \mathbf{w})}[\frac{1}{m}\sum_{k \in \mathcal{S}(\pi, \mathbf{w})}(F_k(\mathbf{w}') - F_k^*)]}{F(\mathbf{w}') - \sum_{k=1}^{K} p_k F_k^*} \geq 0, \tag{5}$$

*which reflects the skew of a client selection strategy* $\pi$. *The first* $\mathbf{w}$ *in* $\rho(\mathcal{S}(\pi, \mathbf{w}), \mathbf{w}')$ *is the parameter vector that governs the client selection and* $\mathbf{w}'$ *is the point at which* $F_k$ *and* $F$ *in the numerator and denominator respectively are evaluated. Note,* $\mathbb{E}_{\mathcal{S}(\pi, \mathbf{w})}[\cdot]$ *is the expectation over the randomness from the selection strategy* $\pi$, *since there can be multiple sets* $\mathcal{S}$ *that* $\pi$ *can map from a specific* $\mathbf{w}$.

*Since $\rho(\mathcal{S}(\pi, \mathbf{w}), \mathbf{w}')$ is a function of versions of the global model $\mathbf{w}$ and $\mathbf{w}'$, which change during training, we define two related metrics that are independent of $\mathbf{w}$ and $\mathbf{w}'$. These metrics enable us to obtain a conservative error bound in the convergence analysis.*

$$\overline{\rho} \triangleq \min_{\mathbf{w}, \mathbf{w}'} \rho(\mathcal{S}(\pi, \mathbf{w}), \mathbf{w}'), \quad \widetilde{\rho} \triangleq \max_{\mathbf{w}} \rho(\mathcal{S}(\pi, \mathbf{w}), \mathbf{w}^*) \tag{6}$$

*where $\mathbf{w}^* = \arg\min_{\mathbf{w}} F(\mathbf{w})$. From (6), we have $\overline{\rho} \leq \widetilde{\rho}$ for any client selection strategy $\pi$.*

**Effect of the Client Selection Strategy on $\overline{\rho}$ and $\widetilde{\rho}$.** For the unbiased client selection strategy $\pi_{\text{rand}}$ we have $\rho(\mathcal{S}(\pi_{\text{rand}}, \mathbf{w}), \mathbf{w}') = 1$ for all $\mathbf{w}$ and $\mathbf{w}'$ since the numerator and denominator of (5) become equal, and $\overline{\rho} = \widetilde{\rho} = 1$. For a client selection strategy $\pi$ that chooses clients with higher $F_k(\mathbf{w})$ more often, $\overline{\rho}$ and $\widetilde{\rho}$ will be larger (and $\geq 1$). In the convergence analysis we show that a larger $\overline{\rho}$ implies faster convergence, albeit with a potential error gap, which is proportional to $(\widetilde{\rho}/\overline{\rho} - 1)$. Motivated by this, in Section 4 we present an adaptive client selection strategy that prefers selecting clients with higher loss $F_k(\mathbf{w})$ and achieves faster convergence speed with low solution bias.

## 3.2 MAIN CONVERGENCE RESULT

Here, we present the convergence results for any client selection strategy $\pi$ for federated averaging with partial device participation in terms of local-global objective gap $\Gamma$, and selection skew $\overline{\rho}, \widetilde{\rho}$.

**Theorem 3.1** (Convergence with Decaying Learning Rate). *Under Assumptions 3.1 to 3.4, for learning rate $\eta_t = \frac{1}{\mu(t+\gamma)}$ with $\gamma = \frac{4L}{\mu}$, and any client selection strategy $\pi$, the error after $T$ iterations of federated averaging with partial device participation satisfies*

$$\mathbb{E}[F(\overline{\mathbf{w}}^{(T)})] - F^* \leq$$

$$\underbrace{\frac{1}{(T+\gamma)}\left[\frac{4L(32\tau^2 G^2 + \sigma^2/m)}{3\mu^2\overline{\rho}} + \frac{8L^2\Gamma}{\mu^2} + \frac{L\gamma\|\overline{\mathbf{w}}^{(0)} - \mathbf{w}^*\|^2}{2}\right]}_{\text{Vanishing Error Term}} + \underbrace{\frac{8L\Gamma}{3\mu}\left(\frac{\widetilde{\rho}}{\overline{\rho}} - 1\right)}_{\text{Non-vanishing bias}, Q(\overline{\rho}, \widetilde{\rho})} \tag{7}$$

To the best of our knowledge, Theorem 3.1 provides the first convergence analysis of federated averaging with a biased client selection strategy $\pi$. We also show the results for fixed learning rate in Appendix A. The proof is presented in Appendix C. The first part of our proof follows techniques presented by Li et al. (2020). Then we introduce the novel concept of selection skew to the proof, and analyze the effect of biased client selection strategies that has not been seen before in previous literature. We highlight that our convergence result is a general analysis that is applicable for any selection strategy $\pi$ that is cognizant of the training progress. In the following paragraphs, we discuss the effects of the two terms in (7) in detail.

**Large $\overline{\rho}$ and Faster Convergence.** A key insight from Theorem 3.1 is that a larger selection skew $\overline{\rho}$ results in faster convergence at the rate $\mathcal{O}(\frac{1}{T\overline{\rho}})$. Note that since we obtain $\overline{\rho}$ (defined in (6)) by taking a minimum of the selection skew $\rho(\mathcal{S}(\pi, \mathbf{w}), \mathbf{w}')$ over $\mathbf{w}, \mathbf{w}'$, this is a conservative bound on the true convergence rate. In practice, since the selection skew $\rho(\mathcal{S}(\pi, \mathbf{w}), \mathbf{w}')$ changes during training depending on the current global model $\mathbf{w}$ and the local models $\mathbf{w}'$, the true convergence rate can be improved by a factor larger than and at least equal to $\overline{\rho}$.

**Non-vanishing Bias Term.** The second term $Q(\overline{\rho}, \widetilde{\rho}) = \frac{8L\Gamma}{3\mu}\left(\frac{\widetilde{\rho}}{\overline{\rho}} - 1\right)$ in (7) denotes the solution bias, which is dependent on the selection strategy. By the definitions of $\overline{\rho}$ and $\widetilde{\rho}$, it follows that $\widetilde{\rho} \geq \overline{\rho}$, which implies that $Q(\overline{\rho}, \widetilde{\rho}) \geq 0$. For an unbiased selection strategy, we have $\overline{\rho} = \widetilde{\rho} = 1$, $Q(\overline{\rho}, \widetilde{\rho}) = 0$, and hence (7) recovers previous bound for unbiased selection strategy as (Li et al., 2020). For $\overline{\rho} > 1$, while we gain faster convergence rate by a factor of $\overline{\rho}$, we cannot guarantee $Q(\overline{\rho}, \widetilde{\rho}) = 0$. Thus, there is a trade-off between the convergence speed and the solution bias. Later in the experimental results, we show that even with biased selection strategies, the term $\frac{\widetilde{\rho}}{\overline{\rho}} - 1$ in $Q(\overline{\rho}, \widetilde{\rho})$ can be close to 0, and hence $Q(\overline{\rho}, \widetilde{\rho})$ has a negligible effect on the final error floor.

## 4 PROPOSED POWER-OF-CHOICE CLIENT SELECTION STRATEGY

From (5) and (6) we discover that a selection strategy $\pi$ that prefers clients with larger $F_k(\mathbf{w}) - F_k^*$ will result in a larger $\overline{\rho}$, yielding faster convergence. Using this insight, a naive client selection strategy can be choosing the clients with highest local loss $F_k(\mathbf{w})$. However, a larger selection skew $\overline{\rho}$ may result in a larger $\overline{\rho}/\widetilde{\rho}$, i.e., a larger non-vanishing error term. This naive selection strategy has another drawback – to find the current local loss $F_k(\mathbf{w})$, it requires sending the current global model to all $K$ clients and having them evaluate $F_k$ and sending it back. This additional communication and computation cost can be prohibitively high because the number of clients $K$ is typically very large, and these clients have limited communication and computation capabilities.

In this section, we use these insights regarding the trade-off between convergence speed, solution bias and communication/computation overhead to propose the POWER-OF-CHOICE client selection strategy. POWER-OF-CHOICE is based on the power of $d$ choices load balancing strategy (Mitzen-macher, 1996), which is extensively used in queueing systems. In the POWER-OF-CHOICE client selection strategy (denoted by $\pi_{\text{pow-d}}$), the central server chooses the active client set $\mathcal{S}^{(t)}$ as follows:

1. **Sample the Candidate Client Set.** The central server samples a candidate set $\mathcal{A}$ of $d$ ($m \leq d \leq K$) clients without replacement such that client $k$ is chosen with probability $p_k$, the fraction of data at the $k$-th client for $k = 1, \ldots K$.

2. **Estimate Local Losses.** The server sends the current global model $\overline{\mathbf{w}}^{(t)}$ to the clients in set $\mathcal{A}$, and these clients compute and send back to the central server their local loss $F_k(\overline{\mathbf{w}}^{(t)})$.

3. **Select Highest Loss Clients.** From the candidate set $\mathcal{A}$, the central server constructs the active client set $\mathcal{S}^{(t)}$ by selecting $m = \max(CK, 1)$ clients with the largest values $F_k(\overline{\mathbf{w}})$, with ties broken at random. These $\mathcal{S}^{(t)}$ clients participate in the training during the next round, consisting of iterations $t + 1, t + 2, \ldots t + \tau$.

**Variations of $\pi_{\text{pow-d}}$.** The three steps of $\pi_{\text{pow-d}}$ can be flexibly modified to take into account practical considerations. For example, intermittent client availability can be accounted for in step 1 by constructing set $\mathcal{A}$ only from the set of available clients in that round. We demonstrate the performance of $\pi_{\text{pow-d}}$ with intermittent client availability in Appendix G.3. The local computation cost and server-client communication cost in step 2 can be reduced or eliminated by the following proposed variants of $\pi_{\text{pow-d}}$ (see Appendix F for their pseudo-codes).

- **Computation-efficient Variant $\pi_{\text{cpow-d}}$:** To save local computation cost, instead of evaluating the $F_k(\mathbf{w})$ by going through the entire local dataset $\mathcal{B}_k$, we use an estimate $\sum_{\xi \in \widehat{\xi}_k} f(\mathbf{w}, \xi)/|\widehat{\xi}_k|$, where $\widehat{\xi}_k$ is the mini-batch of $b$ samples sampled uniformly at random from $\mathcal{B}_k$.

- **Communication- and Computation-efficient Variant $\pi_{\text{rpow-d}}$:** To save both local computation and communication cost, the selected clients for each round sends their accumulated averaged loss over local iterations, i.e., $\frac{1}{\tau |\xi_k^{(l)}|} \sum_{l=t-\tau+1}^{t} \sum_{\xi \in \xi_k^{(l)}} f(\mathbf{w}_k^{(l)}, \xi)$ when they send their local models to the server. The server uses the latest received value from each client as a proxy for $F_k(\mathbf{w})$ to select the clients. For the clients that have not been selected yet, the latest value is set to $\infty$.

- **Adaptive Selection Skew Variant $\pi_{\text{adapow-d}}$:** To minimize the non-vanishing bias term in Theorem 3.1 while simultaneously gaining the benefit of convergence speed from $\overline{\rho}$, we gradually reduce $d$ until $d = m$[1]. This enables convergence speed up in the initial training phase, while eventually diminishing the non-vanishing bias term when $d = m$. Which $d$ to start with and how gradually we decrease $d$ to $m$ is flexible, analogous to setting the environment and hyper-parameters.

**Selection Skew of POWER-OF-CHOICE Strategy.** The size $d$ of the candidate client set $\mathcal{A}$ is an important parameter which controls the trade-off between convergence speed and solution bias. With $d = m$ we have random sampling without replacement in proportion of $p_k$. As $d$ increases, the selection skew $\overline{\rho}$ increases, giving faster error convergence at the risk of a higher error floor. However, note that the convergence analysis replaces $\rho(\mathbf{w}, \mathbf{w}')$ with $\overline{\rho}$ to get a conservative error bound. In

---

[1]$d = m$ makes our proposed POWER-OF-CHOICE strategy to become analogous to an unbiased sampling strategy, which has no non-vanishing bias term.

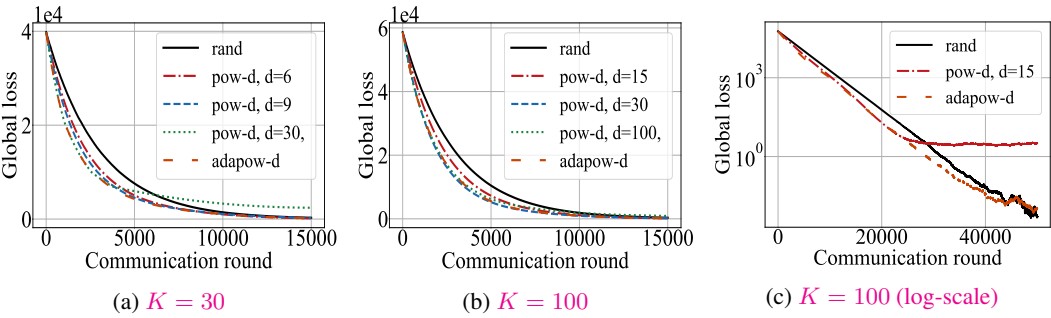

(a) $K = 30$                    (b) $K = 100$                    (c) $K = 100$ (log-scale)

Figure 2: Global loss performance of $\pi_{\text{rand}}$, $\pi_{\text{pow-d}}$, and $\pi_{\text{adapow-d}}$ for the quadratic experiments with $C = 0.1$. $\pi_{\text{pow-d}}$ convergences faster than $\pi_{\text{rand}}$ for even selecting from a small pool of clients ($K = 30$). As convergence speed increases, solution bias also increases for $\pi_{\text{pow-d}}$, but $\pi_{\text{adapow-d}}$ is able to eliminate this solution bias while gaining nearly identical convergence speed to $\pi_{\text{pow-d}}$.

practice, the convergence speed and the solution bias is dictated by $\rho(\overline{\mathbf{w}}^{(\tau\lfloor t/\tau\rfloor)}, \overline{\mathbf{w}}^{(t)})$ which changes during training. With $\pi_{\text{pow-d}}$ which is biased towards higher local losses, we expect the selection skew $\rho(\mathbf{w}, \mathbf{w}')$ to reduce through the course of training. We conjecture that this is why $\pi_{\text{pow-d}}$ gives faster convergence as well as little or no solution bias in our experiments presented in Section 5.

## 5  EXPERIMENTAL RESULTS

We evaluate our proposed $\pi_{\text{pow-d}}$ and its practical variants $\pi_{\text{cpow-d}}$, $\pi_{\text{rpow-d}}$, and $\pi_{\text{adapow-d}}$ by three sets of experiments: (1) quadratic optimization, (2) logistic regression on a synthetic federated dataset, Synthetic(1,1) (Sahu et al., 2019), and (3) DNN trained on a non-iid partitioned FMNIST dataset (Xiao et al., 2017). We also benchmark the selection strategy proposed by Goetz et al. (2019), active federated learning, denoted as $\pi_{\text{afl}}$. Details of the experimental setup are provided in Appendix F, and the code for all experiments are shared in the supplementary material. To validate consistency in our results, we present additional experiments with DNN trained on a non-iid partitioned EMNIST (Cohen et al., 2017) dataset sorted by digits with $K = 500$ clients. We present the results in Appendix G.4.

**Quadratic and Synthetic Simulation Results.** In Figure 2(a), even with few clients ($K = 30$), $\pi_{\text{pow-d}}$ converges faster than $\pi_{\text{rand}}$ with nearly negligible solution bias for small $d$. The convergence speed increases with the increase in $d$, at the cost of higher error floor due to the solution bias. For $K = 100$ in Figure 2(b), $\pi_{\text{pow-d}}$ shows convergence speed-up as with $K = 30$, but the bias is smaller. Figure 3 shows the theoretical values $\overline{\rho}$ and $\widetilde{\rho}/\overline{\rho}$ which represents the convergence speed and the solution bias respectively in our convergence analysis. Compared to $\pi_{\text{rand}}$, $\pi_{\text{pow-d}}$ has higher $\overline{\rho}$ for all $d$ implying higher convergence speed than $\pi_{\text{rand}}$. By varying $d$ we can span different points on the trade-off between the convergence speed and bias. For $d = 15$ and $K = 100$, $\widetilde{\rho}/\overline{\rho}$ of $\pi_{\text{pow-d}}$ and $\pi_{\text{rand}}$ are approximately identical, but $\pi_{\text{pow-d}}$ has higher $\overline{\rho}$, implying that $\pi_{\text{pow-d}}$ can yield higher convergence speed with negligible solution bias. In Appendix G.1, we present

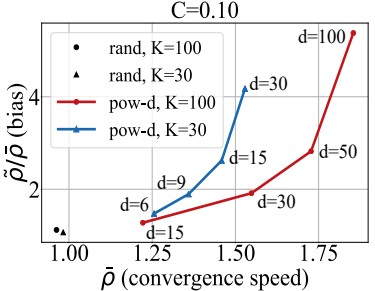

Figure 3: Estimated theoretical values $\overline{\rho}$ and $\overline{\rho}/\widetilde{\rho}$ for the quadratic simulation. The convergence speed ($\overline{\rho}$) and bias ($\widetilde{\rho}/\overline{\rho}$) are consistent with the results shown in Figure 2 for $\pi_{\text{rand}}$ and $\pi_{\text{pow-d}}$.

the clients' selected frequency ratio for $\pi_{\text{pow-d}}$ and $\pi_{\text{rand}}$ which gives novel insights regarding the difference between the two strategies. For the synthetic dataset simulations, we present the global losses in Figure 4 for $\pi_{\text{rand}}$ and $\pi_{\text{pow-d}}$ for different $d$ and $m$. We show that $\pi_{\text{pow-d}}$ converges approximately $3\times$ faster to the global loss $\approx 0.7$ than $\pi_{\text{rand}}$ when $d = 10m$, with a slightly higher error floor. Even with $d = 2m$, we get $2\times$ faster convergence to global loss $\approx 0.7$ than $\pi_{\text{rand}}$.

**Elimination of Selection Skew with $\pi_{\text{adapow-d}}$.** For $\pi_{\text{pow-d}}$, the selection skew is the trade-off for the convergence speed gain in Figure 2 and Figure 4. For both simulations, $\pi_{\text{pow-d}}$ converges slightly above the global minimum value due to the selection skew. We eliminate this selection skew while maintaining the benefit of convergence speed with $\pi_{\text{adapow-d}}$. In Figure 2(a)-(b), $\pi_{\text{adapow-d}}$ shows a

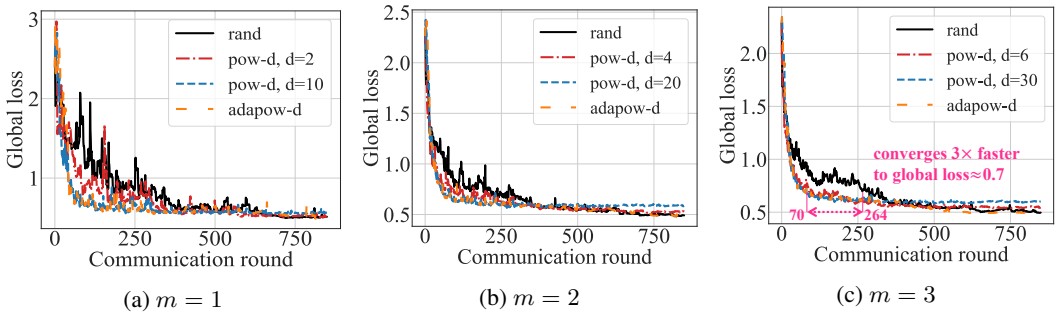

Figure 4: Global loss for logistic regression on the synthetic dataset, Synthetic(1,1), with $\pi_{\text{rand}}$, $\pi_{\text{pow-d}}$, and $\pi_{\text{adapow-d}}$ for $d \in \{2m, 10m\}$ where $K = 30$, $m \in \{1, 2, 3\}$. $\pi_{\text{pow-d}}$ converges approximately $3 \times$ faster for $d = 10m$ and $2 \times$ faster for $d = 2m$ than $\pi_{\text{rand}}$ to the global loss $\approx 0.7$. $\pi_{\text{adapow-d}}$ is able to converge to the minimum global loss $3\times$ faster than $\pi_{\text{rand}}$.

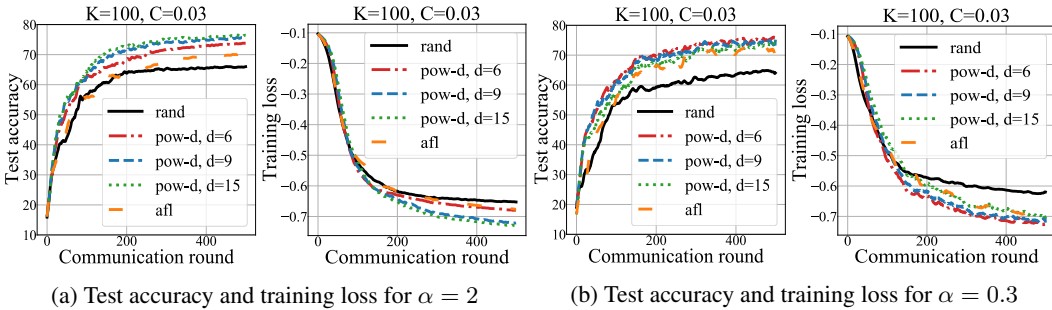

Figure 5: Test accuracy and training loss for different sampling strategies with $K = 100$, $C = 0.03$ for varying $d$ on the FMNIST dataset. For both small and large $\alpha$, $\pi_{\text{pow-d}}$ achieves at least $10\%$ test accuracy improvement than $\pi_{\text{rand}}$ and the training loss converges at a much higher rate than $\pi_{\text{rand}}$.

convergence speed similar to $\pi_{\text{pow-d}}$, $d = K$, but has no selection skew, converging to the same minimum as $\pi_{\text{rand}}$ (see Figure 2(c)). In Figure 4, $\pi_{\text{adapow-d}}$ again shows a convergence speed similar to $\pi_{\text{pow-d}}$, $d = 10m$, but has no adversarial selection skew. As a matter of fact, $\pi_{\text{adapow-d}}$ converges to the minimum global loss value at least $3 \times$ faster than $\pi_{\text{rand}}$. Hence $\pi_{\text{adapow-d}}$ gains the benefit of both worlds from biased client selection: convergence speed and elimination of selection skew.

**Experiments with Heterogeneously Distributed FMNIST.** As elaborated in Appendix F, $\alpha$ determines the data heterogeneity across clients. Smaller $\alpha$ indicates larger data heterogeneity. In Figure 5, we present the test accuracy and training losses for the different sampling strategies from the FMNIST experiments with $\alpha = 0.3$ and $\alpha = 2$. Observe that $\pi_{\text{pow-d}}$ achieves approximately $10\%$ and $5\%$ higher test accuracy than $\pi_{\text{rand}}$ and $\pi_{\text{afl}}$ respectively for both $\alpha = 2$ and $\alpha = 0.3$. For higher $\alpha$ (less data heterogeneity) larger $d$ (more selection skew) performs better than smaller $d$.

Figure 5(a) shows that this performance improvement due to the increase of $d$ eventually converges. For smaller $\alpha$, as in Figure 5(b), smaller $d = 6$ performs better than larger $d$ which shows that too much solution bias is adversarial to the performance in the presence of large data heterogeneity. The observations on training loss are consistent with the test accuracy results.

**Performance of the Communication- and Computation-Efficient variants.** Next, we evaluate $\pi_{\text{cpow-d}}$ and $\pi_{\text{rpow-d}}$ which were introduced in Section 4. In Figure 6, for $\alpha = 2$, $\pi_{\text{rpow-d}}$ and $\pi_{\text{cpow-d}}$ each yields approximately $5\%$ and $6\%$ higher accuracy than $\pi_{\text{rand}}$, but both yield lower accuracy than $\pi_{\text{pow-d}}$ that utilizes the highest computation and communication resources. For $\alpha = 0.3$, $\pi_{\text{cpow-d}}$ and $\pi_{\text{rpow-d}}$ perform as well as $\pi_{\text{pow-d}}$ and give a $10\%$ accuracy improvement over $\pi_{\text{rand}}$. Moreover, $\pi_{\text{pow-d}}$, $\pi_{\text{rpow-d}}$ and $\pi_{\text{cpow-d}}$ all have higher accuracy and faster convergence than $\pi_{\text{afl}}$.

We evaluate the communication and computation efficiency of POWER-OF-CHOICE by comparing different strategies in terms of $R_{60}$, the number of communication rounds required to reach test accuracy $60\%$, and $t_{\text{comp}}$, the average computation time (in seconds) spent per round. The computation time includes the the time taken by the central server to select the clients (including the computation

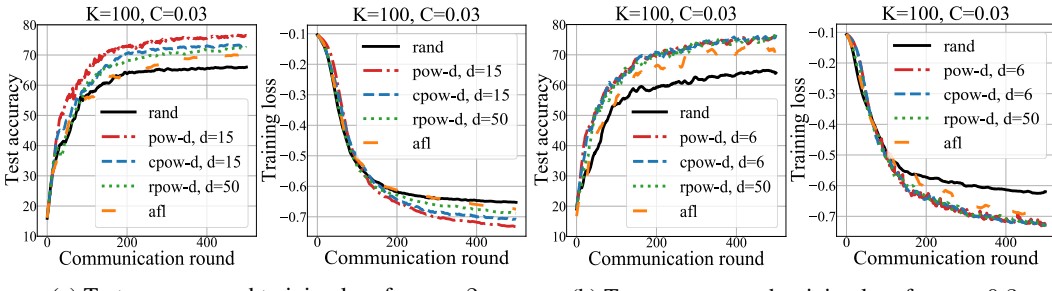

(a) Test accuracy and training loss for $\alpha = 2$      (b) Test accuracy and training loss for $\alpha = 0.3$

Figure 6: Test accuracy and training loss for different sampling strategies including $\pi_{\text{cpow-d}}$ and $\pi_{\text{rpow-d}}$, for $K = 100$, $C = 0.03$ on the FMNIST dataset. $\pi_{\text{rpow-d}}$ which requires no additional communication and minor computation, yields higher test accuracy than $\pi_{\text{rand}}$ and $\pi_{\text{afl}}$.

Table 1: Comparison of $R_{60}$, $t_{\text{comp}}$(sec), and test accuracy (%) for different sampling strategies with $\alpha = 0.3$. In the parentheses we show the ratio of each value with that for $\pi_{\text{rand}}$ with $C = 0.1$.

| | $C = 0.1$ | $C = 0.03$ | | | | |
|---|---|---|---|---|---|---|
| | rand | rand | pow-d, $d = 6$ | cpow-d, $d = 6$ | rpow-d, $d = 50$ | afl |
| $R_{60}$ | 172 | 234(1.36) | 89(0.52) | **80 (0.47)** | **98 (0.57)** | 121(0.70) |
| $t_{\text{comp}}$ | 0.43 | 0.36(0.85) | 0.48(1.13) | **0.37 (0.88)** | **0.37 (0.85)** | 0.36(0.84) |
| Test Acc. | **71.21**±2.41 | 64.87±1.97 | 76.47±0.87 | **76.63**±0.79 | **76.56**±1.00 | 73.28±1.05 |

time for the $d$ clients to compute their local loss values) and the time taken by selected clients to perform local updates. In Table 1, with only $C = 0.03$ fraction of clients, $\pi_{\text{pow-d}}$, $\pi_{\text{cpow-d}}$, and $\pi_{\text{rpow-d}}$ have about $5\%$ higher test accuracy than $(\pi_{\text{rand}}, C = 0.1)$. The $R_{60}$ for $\pi_{\text{pow-d}}$, $\pi_{\text{cpow-d}}$, $\pi_{\text{rpow-d}}$ is $0.52, 0.47, 0.57$ times that of $(\pi_{\text{rand}}, C = 0.1)$ respectively. This implies that even for $\pi_{\text{rpow-d}}$ which does not incur any additional communication cost for client selection, we can get a $2\times$ reduction in the number of communication rounds using $1/3$ of clients compared to $(\pi_{\text{rand}}, C = 0.1)$ and still get higher test accuracy performance. Note that the computation time $t_{\text{comp}}$ for $\pi_{\text{cpow-d}}$ and $\pi_{\text{rpow-d}}$ with $C = 0.03$ is smaller than that of $\pi_{\text{rand}}$ with $C = 0.1$. In Appendix G.2, we show that the results for $\alpha = 2$ are consistent with the $\alpha = 0.3$ case shown in Table 1. In Appendix G.5, we also show that for $C = 0.1$, the results are consistent with the $C = 0.03$ case.

**Effect of Mini-batch Size and Local Epochs**. We evaluate the effect of mini-batch size $b$ and local epochs $\tau$ on the FMNIST experiments with different sets of hyper-parameters: $(b, \tau) \in \{(128, 30), (64, 100)\}$. Note that $(b, \tau) = (64, 30)$ is the default hyper-parameter setting for the previous results. The figures are presented in Appendix G.6. For $b = 128$, we observe that the performance improvement of $\pi_{\text{pow-d}}$ over $\pi_{\text{rand}}$ and $\pi_{\text{afl}}$ is consistent with $b = 64$ (see Figure 12). In Figure 14, for $\tau = 100$, with smaller data heterogeneity, the performance gap between $\pi_{\text{rand}}$ and $\pi_{\text{pow-d}}$ is consistent with that of $\tau = 30$. For larger data heterogeneity, however, increasing the local epochs results in $\pi_{\text{rand}}$ and $\pi_{\text{pow-d}}$ performing similarly. This shows that with larger data heterogeneity, larger $\tau$ results in increasing the selection skew towards specific clients, and weakens generalization.

## 6 CONCLUDING REMARKS

In this work, we present the convergence guarantees for federated learning with partial device participation with any biased client selection strategy. We discover that biasing client selection can speed up the convergence at the rate $\mathcal{O}(\frac{1}{T\bar{\rho}})$ where $\bar{\rho}$ is the selection skew towards clients with higher local losses. Motivated by this insight, we propose the adaptive client selection strategy POWER-OF-CHOICE. Extensive experiments validate that POWER-OF-CHOICE yields $3\times$ faster convergence and $10\%$ higher test accuracy than the baseline federated averaging with random selection. Even with using fewer clients than random selection, POWER-OF-CHOICE converges $2 \times$ faster with high test performance. An interesting future direction is to improve the fairness (Li et al., 2019; Yu et al., 2020; Lyu et al., 2020; Mohri et al., 2019) and robustness (Pillutla et al., 2019) of the POWER-OF-CHOICE strategy by modifying step 3 of the POWER-OF-CHOICE algorithm to use a different metric such as the clipped loss or the $q$-fair loss proposed Li et al. (2019) instead of $F_k(\mathbf{w})$.

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

## A    ADDITIONAL THEOREM

**Theorem A.1** (Convergence with Fixed Learning Rate). *Under Assumptions 3.1 to 3.4, a fixed learning rate $\eta \leq \min\{\frac{1}{2\mu B}, \frac{1}{4L}\}$ where $B = 1 + \frac{3\overline{\rho}}{8}$, and any client selection strategy $\pi$ as defined above, the error after $T$ iterations of federated averaging with partial device participation satisfies*

$$F(\overline{\mathbf{w}}^{(T)}) - F^*$$

$$\leq \underbrace{\frac{L}{\mu}\left[1 - \eta\mu\left(1 + \frac{3\overline{\rho}}{8}\right)\right]^T \left(F(\overline{\mathbf{w}}^{(0)}) - F^* - \frac{4\left[\eta\left(32\tau^2 G^2 + \frac{\sigma^2}{m} + 6\overline{\rho}L\Gamma\right) + 2\Gamma(\widetilde{\rho} - \overline{\rho})\right]}{8 + 3\overline{\rho}}\right)}_{\text{Vanishing Term}}$$

$$+ \underbrace{\frac{4L\eta\left(32\tau^2 G^2 + \frac{\sigma^2}{m} + 6\overline{\rho}L\Gamma\right)}{\mu(8 + 3\overline{\rho})} + \frac{8L\Gamma(\widetilde{\rho} - \overline{\rho})}{\mu(8 + 3\overline{\rho})}}_{\text{Non-vanishing bias}} \tag{8}$$

As $T \to \infty$ the first term in (8) goes to 0 and the second term becomes the bias term for the fixed learning rate case. For a small $\eta$, we have that the bias term for the fixed learning rate case in Theorem A.1 is upper bounded by $\frac{8L\Gamma}{3\mu}\left(\frac{\widetilde{\rho}}{\overline{\rho}} - 1\right)$ which is identical to the decaying-learning rate case. The proof is presented in Appendix D.

## B    PRELIMINARIES FOR PROOF OF THEOREM 3.1 AND THEOREM A.1

We present the preliminary lemmas used for proof of Theorem 3.1 and Theorem A.1. We will denote the expectation over the sampling random source $\mathcal{S}^{(t)}$ as $\mathbb{E}_{\mathcal{S}^{(t)}}$ and the expectation over all the random sources as $\mathbb{E}$.

**Lemma B.1.** *Suppose $F_k$ is $L-$smooth with global minimum at $\mathbf{w}_k^*$, then for any $\mathbf{w}_k$ in the domain of $F_k$, we have that*

$$\|\nabla F_k(\mathbf{w}_k)\|^2 \leq 2L(F_k(\mathbf{w}_k) - F_k(\mathbf{w}_k^*)) \tag{9}$$

*Proof.*

$$F_k(\mathbf{w}_k) - F_k(\mathbf{w}_k^*) - \langle\nabla F_k(\mathbf{w}_k^*), \mathbf{w}_k - \mathbf{w}_k^*\rangle \geq \frac{1}{2L}\|\nabla F_k(\mathbf{w}_k) - \nabla F_k(\mathbf{w}_k^*)\|^2 \tag{10}$$

$$F_k(\mathbf{w}_k) - F_k(\mathbf{w}_k^*) \geq \frac{1}{2L}\|\nabla F_k(\mathbf{w}_k)\|^2 \tag{11}$$

$\square$

**Lemma B.2** (Expected average discrepancy between $\overline{\mathbf{w}}^{(t)}$ and $\mathbf{w}_k^{(t)}$ for $k \in \mathcal{S}^{(t)}$)**.**

$$\frac{1}{m}\mathbb{E}[\sum_{k\in\mathcal{S}^{(t)}}\|\overline{\mathbf{w}}^{(t)} - \mathbf{w}_k^{(t)}\|^2] \leq 16\eta_t^2\tau^2 G^2 \tag{12}$$

*Proof.*

$$\frac{1}{m}\sum_{k\in\mathcal{S}^{(t)}}\|\overline{\mathbf{w}}^{(t)} - \mathbf{w}_k^{(t)}\|^2 = \frac{1}{m}\sum_{k\in\mathcal{S}^{(t)}}\|\frac{1}{m}\sum_{k'\in\mathcal{S}^{(t)}}(\mathbf{w}_{k'}^{(t)} - \mathbf{w}_k^{(t)})\|^2 \tag{13}$$

$$\leq \frac{1}{m^2}\sum_{k\in\mathcal{S}^{(t)}}\sum_{k'\in\mathcal{S}^{(t)}}\|\mathbf{w}_{k'}^{(t)} - \mathbf{w}_k^{(t)}\|^2 \tag{14}$$

$$= \frac{1}{m^2}\sum_{\substack{k\neq k',\\ k,k'\in\mathcal{S}^{(t)}}}\|\mathbf{w}_{k'}^{(t)} - \mathbf{w}_k^{(t)}\|^2 \tag{15}$$

Observe from the update rule that $k$, $k'$ are in the same set $\mathcal{S}^{(t)}$ and hence the terms where $k = k'$ in the summation in (14) will be zero resulting in (15). Moreover for any arbitrary $t$ there is a $t_0$ such that $0 \leq t - t_0 < \tau$ that $\mathbf{w}_{k'}^{(t_0)} = \mathbf{w}_k^{(t_0)}$ since the selected clients are updated with the global model at every $\tau$. Hence even for an arbitrary $t$ we have that the difference between $\|\mathbf{w}_{k'}^{(t)} - \mathbf{w}_k^{(t)}\|^2$ is upper bounded by $\tau$ updates. With non-increasing $\eta_t$ over $t$ and $\eta_{t_0} \leq 2\eta_t$, (15) can be further bounded as,

$$\frac{1}{m^2} \sum_{\substack{k \neq k', \\ k,k' \in \mathcal{S}^{(t)}}} \|\mathbf{w}_{k'}^{(t)} - \mathbf{w}_k^{(t)}\|^2 \leq \frac{1}{m^2} \sum_{\substack{k \neq k', \\ k,k' \in \mathcal{S}^{(t)}}} \|\sum_{i=t_0}^{t_0+\tau-1} \eta_i (g_{k'}(\mathbf{w}_{k'}^{(i)}, \xi_{k'}^{(i)}) - g_k(\mathbf{w}_k^{(i)}, \xi_k^{(i)}))\|^2 \quad (16)$$

$$\leq \frac{\eta_{t_0}^2 \tau}{m^2} \sum_{\substack{k \neq k', \\ k,k' \in \mathcal{S}^{(t)}}} \sum_{i=t_0}^{t_0+\tau-1} \|(g_{k'}(\mathbf{w}_{k'}^{(i)}, \xi_{k'}^{(i)}) - g_k(\mathbf{w}_k^{(i)}, \xi_k^{(i)}))\|^2 \quad (17)$$

$$\leq \frac{\eta_{t_0}^2 \tau}{m^2} \sum_{\substack{k \neq k', \\ k,k' \in \mathcal{S}^{(t)}}} \sum_{i=t_0}^{t_0+\tau-1} [2\|g_{k'}(\mathbf{w}_{k'}^{(i)}, \xi_{k'}^{(i)})\|^2 + 2\|g_k(\mathbf{w}_k^{(i)}, \xi_k^{(i)})\|^2] \quad (18)$$

By taking expectation over (18),

$$\mathbb{E}[\frac{1}{m^2} \sum_{\substack{k \neq k', \\ k,k' \in \mathcal{S}^{(t)}}} \|\mathbf{w}_{k'}^{(t)} - \mathbf{w}_k^{(t)}\|^2] \leq \frac{2\eta_{t_0}^2 \tau}{m^2} \mathbb{E}[\sum_{\substack{k \neq k', \\ k,k' \in \mathcal{S}^{(t)}}} \sum_{i=t_0}^{t_0+\tau-1} (\|g_{k'}(\mathbf{w}_{k'}^{(i)}, \xi_{k'}^{(i)})\|^2 + \|g_k(\mathbf{w}_k^{(i)}, \xi_k^{(i)})\|^2)]$$

$$(19)$$

$$\leq \frac{2\eta_{t_0}^2 \tau}{m^2} \mathbb{E}_{\mathcal{S}^{(t)}}[\sum_{\substack{k \neq k', \\ k,k' \in \mathcal{S}^{(t)}}} \sum_{i=t_0}^{t_0+\tau-1} 2G^2] \quad (20)$$

$$= \frac{2\eta_{t_0}^2 \tau}{m^2} \mathbb{E}_{\mathcal{S}^{(t)}}[\sum_{\substack{k \neq k', \\ k,k' \in \mathcal{S}^{(t)}}} 2\tau G^2] \quad (21)$$

$$\leq \frac{16\eta_t^2 (m-1) \tau^2 G^2}{m} \quad (22)$$

$$\leq 16\eta_t^2 \tau^2 G^2 \quad (23)$$

where (22) is because there can be at most $m(m-1)$ pairs such that $k \neq k'$ in $\mathcal{S}^{(t)}$. $\qquad\square$

**Lemma B.3** (Upper bound for expectation over $\|\overline{\mathbf{w}}^{(t)} - \mathbf{w}^*\|^2$ for any selection strategy $\pi$). *With $\mathbb{E}[\cdot]$, the total expectation over all random sources including the random source from selection strategy we have the upper bound:*

$$\mathbb{E}[\|\overline{\mathbf{w}}^{(t)} - \mathbf{w}^*\|^2] \leq \frac{1}{m} \mathbb{E}[\sum_{k \in \mathcal{S}^{(t)}} \|\mathbf{w}_k^{(t)} - \mathbf{w}^*\|^2] \quad (24)$$

*Proof.*

$$\mathbb{E}[\|\overline{\mathbf{w}}^{(t)} - \mathbf{w}^*\|^2] = \mathbb{E}[\|\frac{1}{m} \sum_{k \in \mathcal{S}^{(t)}} \mathbf{w}_k^{(t)} - \mathbf{w}^*\|^2] = \mathbb{E}[\|\frac{1}{m} \sum_{k \in \mathcal{S}^{(t)}} (\mathbf{w}_k^{(t)} - \mathbf{w}^*)\|^2] \quad (25)$$

$$\leq \frac{1}{m} \mathbb{E}[\sum_{k \in \mathcal{S}^{(t)}} \|\mathbf{w}_k^{(t)} - \mathbf{w}^*\|^2] \quad (26)$$

$\qquad\square$

# C   PROOF OF THEOREM 3.1

With $\overline{\mathbf{g}}^{(t)} = \frac{1}{m} \sum_{k \in \mathcal{S}^{(t)}} g_k(\mathbf{w}_k^{(t)}, \xi_k^{(t)})$ as defined in Section 2, we have that

$$\|\overline{\mathbf{w}}^{(t+1)} - \mathbf{w}^*\|^2 = \|\overline{\mathbf{w}}^{(t)} - \eta_t \overline{\mathbf{g}}^{(t)} - \mathbf{w}^*\|^2 \tag{27}$$

$$= \|\overline{\mathbf{w}}^{(t)} - \eta_t \overline{\mathbf{g}}^{(t)} - \mathbf{w}^* - \frac{\eta_t}{m} \sum_{k \in \mathcal{S}^{(t)}} \nabla F_k(\mathbf{w}_k^{(t)}) + \frac{\eta_t}{m} \sum_{k \in \mathcal{S}^{(t)}} \nabla F_k(\mathbf{w}_k^{(t)})\|^2 \tag{28}$$

$$= \|\overline{\mathbf{w}}^{(t)} - \mathbf{w}^* - \frac{\eta_t}{m} \sum_{k \in \mathcal{S}^{(t)}} \nabla F_k(\mathbf{w}_k^{(t)})\|^2 + \eta_t^2 \|\frac{1}{m} \sum_{k \in \mathcal{S}^{(t)}} \nabla F_k(\mathbf{w}_k^{(t)}) - \overline{\mathbf{g}}^{(t)}\|^2$$

$$+ 2\eta_t \langle \overline{\mathbf{w}}^{(t)} - \mathbf{w}^* - \frac{\eta_t}{m} \sum_{k \in \mathcal{S}^{(t)}} \nabla F_k(\mathbf{w}_k^{(t)}), \frac{1}{m} \sum_{k \in \mathcal{S}^{(t)}} \nabla F_k(\mathbf{w}_k^{(t)}) - \overline{\mathbf{g}}^{(t)} \rangle \tag{29}$$

$$= \|\overline{\mathbf{w}}^{(t)} - \mathbf{w}^*\|^2 \underbrace{- 2\eta_t \langle \overline{\mathbf{w}}^{(t)} - \mathbf{w}^*, \frac{1}{m} \sum_{k \in \mathcal{S}^{(t)}} \nabla F_k(\mathbf{w}_k^{(t)}) \rangle}_{A_1}$$

$$\underbrace{+ 2\eta_t \langle \overline{\mathbf{w}}^{(t)} - \mathbf{w}^* - \frac{\eta_t}{m} \sum_{k \in \mathcal{S}^{(t)}} \nabla F_k(\mathbf{w}_k^{(t)}), \frac{1}{m} \sum_{k \in \mathcal{S}^{(t)}} \nabla F_k(\mathbf{w}_k^{(t)}) - \overline{\mathbf{g}}^{(t)} \rangle}_{A_2}$$

$$\underbrace{+ \eta_t^2 \|\frac{1}{m} \sum_{k \in \mathcal{S}^{(t)}} \nabla F_k(\mathbf{w}_k^{(t)})\|^2}_{A_3} + \underbrace{\eta_t^2 \|\frac{1}{m} \sum_{k \in \mathcal{S}^{(t)}} \nabla F_k(\mathbf{w}_k^{(t)}) - \overline{\mathbf{g}}^{(t)}\|^2}_{A_4} \tag{30}$$

First let's bound $A_1$.

$$- 2\eta_t \langle \overline{\mathbf{w}}^{(t)} - \mathbf{w}^*, \frac{1}{m} \sum_{k \in \mathcal{S}^{(t)}} \nabla F_k(\mathbf{w}_k^{(t)}) \rangle = -\frac{2\eta_t}{m} \sum_{k \in \mathcal{S}^{(t)}} \langle \overline{\mathbf{w}}^{(t)} - \mathbf{w}^*, \nabla F_k(\mathbf{w}_k^{(t)}) \rangle \tag{31}$$

$$= -\frac{2\eta_t}{m} \sum_{k \in \mathcal{S}^{(t)}} \langle \overline{\mathbf{w}}^{(t)} - \mathbf{w}_k^{(t)}, \nabla F_k(\mathbf{w}_k^{(t)}) \rangle - \frac{2\eta_t}{m} \sum_{k \in \mathcal{S}^{(t)}} \langle \mathbf{w}_k^{(t)} - \mathbf{w}^*, \nabla F_k(\mathbf{w}_k^{(t)}) \rangle \tag{32}$$

$$\leq \frac{\eta_t}{m} \sum_{k \in \mathcal{S}^{(t)}} \left( \frac{1}{\eta_t} \|\overline{\mathbf{w}}^{(t)} - \mathbf{w}_k^{(t)}\|^2 + \eta_t \|\nabla F_k(\mathbf{w}_k^{(t)})\|^2 \right) - \frac{2\eta_t}{m} \sum_{k \in \mathcal{S}^{(t)}} \langle \mathbf{w}_k^{(t)} - \mathbf{w}^*, \nabla F_k(\mathbf{w}_k^{(t)}) \rangle \tag{33}$$

$$= \frac{1}{m} \sum_{k \in \mathcal{S}^{(t)}} \|\overline{\mathbf{w}}^{(t)} - \mathbf{w}_k^{(t)}\|^2 + \frac{\eta_t^2}{m} \sum_{k \in \mathcal{S}^{(t)}} \|\nabla F_k(\mathbf{w}_k^{(t)})\|^2 - \frac{2\eta_t}{m} \sum_{k \in \mathcal{S}^{(t)}} \langle \mathbf{w}_k^{(t)} - \mathbf{w}^*, \nabla F_k(\mathbf{w}_k^{(t)}) \rangle \tag{34}$$

$$\leq \frac{1}{m} \sum_{k \in \mathcal{S}^{(t)}} \|\overline{\mathbf{w}}^{(t)} - \mathbf{w}_k^{(t)}\|^2 + \frac{2L\eta_t^2}{m} \sum_{k \in \mathcal{S}^{(t)}} (F_k(\mathbf{w}_k^{(t)}) - F_k^*)$$

$$- \frac{2\eta_t}{m} \sum_{k \in \mathcal{S}^{(t)}} \langle \mathbf{w}_k^{(t)} - \mathbf{w}^*, \nabla F_k(\mathbf{w}_k^{(t)}) \rangle \tag{35}$$

$$\leq \frac{1}{m} \sum_{k \in \mathcal{S}^{(t)}} \|\overline{\mathbf{w}}^{(t)} - \mathbf{w}_k^{(t)}\|^2 + \frac{2L\eta_t^2}{m} \sum_{k \in \mathcal{S}^{(t)}} (F_k(\mathbf{w}_k^{(t)}) - F_k^*)$$

$$- \frac{2\eta_t}{m} \sum_{k \in \mathcal{S}^{(t)}} \left[ (F_k(\mathbf{w}_k^{(t)}) - F_k(\mathbf{w}^*)) + \frac{\mu}{2} \|\mathbf{w}_k^{(t)} - \mathbf{w}^*\|^2 \right] \tag{36}$$

$$\leq 16\eta_t^2 \tau^2 G^2 - \frac{\eta_t \mu}{m} \sum_{k \in \mathcal{S}^{(t)}} \|\mathbf{w}_k^{(t)} - \mathbf{w}^*\|^2 + \frac{2L\eta_t^2}{m} \sum_{k \in \mathcal{S}^{(t)}} (F_k(\mathbf{w}_k^{(t)}) - F_k^*)$$

$$- \frac{2\eta_t}{m} \sum_{k \in \mathcal{S}^{(t)}} (F_k(\mathbf{w}_k^{(t)}) - F_k(\mathbf{w}^*)) \tag{37}$$

where (33) is due to the AM-GM inequality and Cauchy–Schwarz inequality, (35) is due to Lemma B.1, (36) is due to the $\mu$-convexity of $F_k$, and (37) is due to Lemma B.2. Next, in expectation, $\mathbb{E}[A_2] = 0$ due to the unbiased gradient. Next again with Lemma B.1 we bound $A_3$ as follows:

$$\eta_t^2 \| \frac{1}{m} \sum_{k \in \mathcal{S}^{(t)}} \nabla F_k(\mathbf{w}_k^{(t)}) \|^2 = \frac{\eta_t^2}{m} \sum_{k \in \mathcal{S}^{(t)}} \left\| \nabla F_k(\mathbf{w}_k^{(t)}) \right\|^2 \tag{38}$$

$$\leq \frac{2L\eta_t^2}{m} \sum_{k \in \mathcal{S}^{(t)}} (F_k(\mathbf{w}_k^{(t)}) - F_k^*) \tag{39}$$

Lastly we can bound $A_4$ using the bound of variance of stochastic gradients as,

$$\mathbb{E}[\eta_t^2 \| \frac{1}{m} \sum_{k \in \mathcal{S}^{(t)}} \nabla F_k(\mathbf{w}_k^{(t)}) - \overline{\mathbf{g}}^{(t)} \|^2] = \eta_t^2 \mathbb{E}[\| \sum_{k \in \mathcal{S}^{(t)}} \frac{1}{m} (g_k(\mathbf{w}_k^{(t)}, \xi_k^{(t)}) - \nabla F_k(\mathbf{w}_k^{(t)})) \|^2] \tag{40}$$

$$= \frac{\eta_t^2}{m^2} \mathbb{E}_{\mathcal{S}^{(t)}} [\sum_{k \in \mathcal{S}^{(t)}} \mathbb{E}\|g_k(\mathbf{w}_k^{(t)}, \xi_k^{(t)}) - \nabla F_k(\mathbf{w}_k^{(t)})\|^2] \tag{41}$$

$$\leq \frac{\eta_t^2 \sigma^2}{m} \tag{42}$$

Using the bounds of $A_1, A_2, A_3, A_4$ above we have that the expectation of the LHS of (27) is bounded as

$$\mathbb{E}[\|\overline{\mathbf{w}}^{(t+1)} - \mathbf{w}^*\|^2]$$
$$\leq \mathbb{E}[\|\overline{\mathbf{w}}^{(t)} - \mathbf{w}^*\|^2] - \frac{\eta_t \mu}{m} \mathbb{E}[\sum_{k \in \mathcal{S}^{(t)}} \|\mathbf{w}_k^{(t)} - \mathbf{w}^*\|^2] + 16\eta_t^2 \tau^2 G^2$$
$$+ \frac{\eta_t^2 \sigma^2}{m} + \frac{4L\eta_t^2}{m} \mathbb{E}[\sum_{k \in \mathcal{S}^{(t)}} (F_k(\mathbf{w}_k^{(t)}) - F_k^*)] - \frac{2\eta_t}{m} \mathbb{E}[\sum_{k \in \mathcal{S}^{(t)}} (F_k(\mathbf{w}_k^{(t)}) - F_k(\mathbf{w}^*))] \tag{43}$$
$$\leq (1 - \eta_t \mu) \mathbb{E}[\|\overline{\mathbf{w}}^{(t)} - \mathbf{w}^*\|^2] + 16\eta_t^2 \tau^2 G^2$$
$$+ \underbrace{\frac{\eta_t^2 \sigma^2}{m} + \frac{4L\eta_t^2}{m} \mathbb{E}[\sum_{k \in \mathcal{S}^{(t)}} (F_k(\mathbf{w}_k^{(t)}) - F_k^*)] - \frac{2\eta_t}{m} \mathbb{E}[\sum_{k \in \mathcal{S}^{(t)}} (F_k(\mathbf{w}_k^{(t)}) - F_k(\mathbf{w}^*))]}_{A_5} \tag{44}$$

where (44) is due to Lemma B.3. Now we aim to bound $A_5$ in (44). First we can represent $A_5$ in a different form as:

$$\mathbb{E}[\frac{4L\eta_t^2}{m} \sum_{k \in \mathcal{S}^{(t)}} (F_k(\mathbf{w}_k^{(t)}) - F_k^*) - \frac{2\eta_t}{m} \sum_{k \in \mathcal{S}^{(t)}} (F_k(\mathbf{w}_k^{(t)}) - F_k(\mathbf{w}^*))]$$

$$= \mathbb{E}[\frac{4L\eta_t^2}{m} \sum_{k \in \mathcal{S}^{(t)}} F_k(\mathbf{w}_k^{(t)}) - \frac{2\eta_t}{m} \sum_{k \in \mathcal{S}^{(t)}} F_k(\mathbf{w}_k^{(t)}) - \frac{2\eta_t}{m} \sum_{k \in \mathcal{S}^{(t)}} (F_k^* - F_k(\mathbf{w}^*))$$

$$+ \frac{2\eta_t}{m} \sum_{k \in \mathcal{S}^{(t)}} F_k^* - \frac{4L\eta_t^2}{m} \sum_{k \in \mathcal{S}^{(t)}} F_k^*] \tag{45}$$

$$= \mathbb{E}[\underbrace{\frac{2\eta_t(2L\eta_t - 1)}{m} \sum_{k \in \mathcal{S}^{(t)}} (F_k(\mathbf{w}_k^{(t)}) - F_k^*)}_{A_6}] + 2\eta_t \mathbb{E}[\frac{1}{m} \sum_{k \in \mathcal{S}^{(t)}} (F_k(\mathbf{w}^*) - F_k^*)] \tag{46}$$

Now with $\eta_t < 1/(4L)$ and $\nu_t = 2\eta_t(1 - 2L\eta_t)$, we have that $A_6$ can be rewritten and bounded as

$$-\frac{\nu_t}{m}\sum_{k\in\mathcal{S}^{(t)}}(F_k(\mathbf{w}_k^{(t)}) - F_k(\overline{\mathbf{w}}^{(t)}) + F_k(\overline{\mathbf{w}}^{(t)}) - F_k^*)$$

$$= -\frac{\nu_t}{m}\sum_{k\in\mathcal{S}^{(t)}}(F_k(\mathbf{w}_k^{(t)}) - F_k(\overline{\mathbf{w}}^{(t)})) - \frac{\nu_t}{m}\sum_{k\in\mathcal{S}^{(t)}}(F_k(\overline{\mathbf{w}}^{(t)}) - F_k^*) \tag{47}$$

$$\leq -\frac{\nu_t}{m}\sum_{k\in\mathcal{S}^{(t)}}\left[\langle\nabla F_k(\overline{\mathbf{w}}^{(t)}), \mathbf{w}_k^{(t)} - \overline{\mathbf{w}}^{(t)}\rangle + \frac{\mu}{2}\|\mathbf{w}_k^{(t)} - \overline{\mathbf{w}}^{(t)}\|^2\right] - \frac{\nu_t}{m}\sum_{k\in\mathcal{S}^{(t)}}(F_k(\overline{\mathbf{w}}^{(t)}) - F_k^*) \tag{48}$$

$$\leq \frac{\nu_t}{m}\sum_{k\in\mathcal{S}^{(t)}}\left[\eta_t L(F_k(\overline{\mathbf{w}}^{(t)}) - F_k^*) + \left(\frac{1}{2\eta_t} - \frac{\mu}{2}\right)\|\mathbf{w}_k^{(t)} - \overline{\mathbf{w}}^{(t)}\|^2\right] - \frac{\nu_t}{m}\sum_{k\in\mathcal{S}^{(t)}}(F_k(\overline{\mathbf{w}}^{(t)}) - F_k^*) \tag{49}$$

$$= -\frac{\nu_t}{m}(1 - \eta_t L)\sum_{k\in\mathcal{S}^{(t)}}(F_k(\overline{\mathbf{w}}^{(t)}) - F_k^*) + \left(\frac{\nu_t}{2\eta_t m} - \frac{\nu_t\mu}{2m}\right)\sum_{k\in\mathcal{S}^{(t)}}\|\mathbf{w}_k^{(t)} - \overline{\mathbf{w}}^{(t)}\|^2 \tag{50}$$

$$\leq -\frac{\nu_t}{m}(1 - \eta_t L)\sum_{k\in\mathcal{S}^{(t)}}(F_k(\overline{\mathbf{w}}^{(t)}) - F_k^*) + \frac{1}{m}\sum_{k\in\mathcal{S}^{(t)}}\|\mathbf{w}_k^{(t)} - \overline{\mathbf{w}}^{(t)}\|^2 \tag{51}$$

where (48) is due to $\mu-$convexity, (49) is due to Lemma B.1 and the AM-GM inequality and Cauchy–Schwarz inequality, and (51) is due to the fact that $\frac{\nu_t(1-\eta_t\mu)}{2\eta_t} \leq 1$. Hence using this bound of $A_6$ we can upper bound $A_5$ as

$$\mathbb{E}[\frac{4L\eta_t^2}{m}\sum_{k\in\mathcal{S}^{(t)}}(F_k(\mathbf{w}_k^{(t)}) - F_k^*) - \frac{2\eta_t}{m}\sum_{k\in\mathcal{S}^{(t)}}(F_k(\mathbf{w}_k^{(t)}) - F_k(\mathbf{w}^*))]$$

$$\leq \frac{1}{m}\mathbb{E}[\sum_{k\in\mathcal{S}^{(t)}}\|\mathbf{w}_k^{(t)} - \overline{\mathbf{w}}^{(t)}\|^2] - \frac{\nu_t}{m}(1 - \eta_t L)\mathbb{E}[\sum_{k\in\mathcal{S}^{(t)}}(F_k(\overline{\mathbf{w}}^{(t)}) - F_k^*)]$$

$$+ \frac{2\eta_t}{m}\mathbb{E}[\sum_{k\in\mathcal{S}^{(t)}}(F_k(\mathbf{w}^*) - F_k^*)] \tag{52}$$

$$\leq 16\eta_t^2\tau^2 G^2 - \frac{\nu_t}{m}(1 - \eta_t L)\mathbb{E}[\sum_{k\in\mathcal{S}^{(t)}}(F_k(\overline{\mathbf{w}}^{(t)}) - F_k^*)] + \frac{2\eta_t}{m}\mathbb{E}[\sum_{k\in\mathcal{S}^{(t)}}(F_k(\mathbf{w}^*) - F_k^*)] \tag{53}$$

$$= 16\eta_t^2\tau^2 G^2 - \nu_t(1 - \eta_t L)\mathbb{E}[\rho(\mathcal{S}(\pi, \overline{\mathbf{w}}^{(\tau\lfloor t/\tau\rfloor)}), \overline{\mathbf{w}}^{(t)})(F(\overline{\mathbf{w}}^{(t)}) - \sum_{k=1}^{K}p_k F_k^*)]$$

$$+ 2\eta_t\mathbb{E}[\rho(\mathcal{S}(\pi, \overline{\mathbf{w}}^{(\tau\lfloor t/\tau\rfloor)}), \mathbf{w}^*)(F^* - \sum_{k=1}^{K}p_k F_k^*)] \tag{54}$$

$$\leq 16\eta_t^2\tau^2 G^2 \underbrace{-\nu_t(1 - \eta_t L)\overline{\rho}(\mathbb{E}[F(\overline{\mathbf{w}}^{(t)})] - \sum_{k=1}^{K}p_k F_k^*) + 2\eta_t\widetilde{\rho}\Gamma}_{A_7} \tag{55}$$

where (54) is due to the definition of $\rho(\mathcal{S}(\pi, \mathbf{w}), \mathbf{w}')$ in Definition 3.2 and (55) is due to the definition of $\Gamma$ in Definition 3.1 and the definitions of $\overline{\rho}$, $\widetilde{\rho}$ in Definition 3.2. We can expand $A_7$ in (55) as

$$- \nu_t(1 - \eta_t L)\overline{\rho}(\mathbb{E}[F(\overline{\mathbf{w}}^{(t)})] - \sum_{k=1}^{K} p_k F_k^*) \tag{56}$$

$$= - \nu_t(1 - \eta_t L)\overline{\rho} \sum_{k=1}^{K} p_k(\mathbb{E}[F_k(\overline{\mathbf{w}}^{(t)}) - F^* + F^* - F_k^*) \tag{57}$$

$$= - \nu_t(1 - \eta_t L)\overline{\rho} \sum_{k=1}^{K} p_k(\mathbb{E}[F_k(\overline{\mathbf{w}}^{(t)}] - F^*) - \nu_t(1 - \eta_t L)\overline{\rho} \sum_{k=1}^{K} p_k(F^* - F_k^*) \tag{58}$$

$$= - \nu_t(1 - \eta_t L)\overline{\rho}(\mathbb{E}[F(\overline{\mathbf{w}}^{(t)})] - F^*) - \nu_t(1 - \eta_t L)\overline{\rho}\Gamma \tag{59}$$

$$\leq - \frac{\nu_t(1 - \eta_t L)\mu\overline{\rho}}{2}\mathbb{E}[\|\overline{\mathbf{w}}^{(t)} - \mathbf{w}^*\|^2] - \nu_t(1 - \eta_t L)\overline{\rho}\Gamma \tag{60}$$

$$\leq - \frac{3\eta_t\mu\overline{\rho}}{8}\mathbb{E}[\|\overline{\mathbf{w}}^{(t)} - \mathbf{w}^*\|^2] - 2\eta_t(1 - 2L\eta_t)(1 - \eta_t L)\overline{\rho}\Gamma \tag{61}$$

$$\leq - \frac{3\eta_t\mu\overline{\rho}}{8}\mathbb{E}[\|\overline{\mathbf{w}}^{(t)} - \mathbf{w}^*\|^2] - 2\eta_t\overline{\rho}\Gamma + 6\eta_t^2\overline{\rho}L\Gamma \tag{62}$$

where (60) is due to the $\mu-$convexity, (61) is due to $-2\eta_t(1 - 2L\eta_t)(1 - \eta_t L) \leq -\frac{3}{4}\eta_t$, and (62) is due to $-(1 - 2L\eta_t)(1 - \eta_t L) \leq -(1 - 3L\eta_t)$. Hence we can finally bound $A_5$ as

$$\frac{4L\eta_t^2}{m}\mathbb{E}[\sum_{k \in \mathcal{S}^{(t)}} (F_k(\mathbf{w}_k^{(t)}) - F_k^*) - \frac{2\eta_t}{m} \sum_{k \in \mathcal{S}^{(t)}} (F_k(\mathbf{w}_k^{(t)}) - F_k(\mathbf{w}^*))]$$

$$\leq - \frac{3\eta_t\mu\overline{\rho}}{8}\mathbb{E}[\|\overline{\mathbf{w}}^{(t)} - \mathbf{w}^*\|^2] + 2\eta_t\Gamma(\widetilde{\rho} - \overline{\rho}) + \eta_t^2(6\overline{\rho}L\Gamma + 16\tau^2 G^2) \tag{63}$$

Now we can bound $\mathbb{E}[\|\overline{\mathbf{w}}^{(t+1)} - \mathbf{w}^*\|^2]$ as

$$\mathbb{E}[\|\overline{\mathbf{w}}^{(t+1)} - \mathbf{w}^*\|^2] \leq \left[1 - \eta_t\mu\left(1 + \frac{3\overline{\rho}}{8}\right)\right]\mathbb{E}[\|\overline{\mathbf{w}}^{(t)} - \mathbf{w}^*\|^2]$$

$$+ \eta_t^2\left(32\tau^2 G^2 + \frac{\sigma^2}{m} + 6\overline{\rho}L\Gamma\right) + 2\eta_t\Gamma(\widetilde{\rho} - \overline{\rho}) \tag{64}$$

By defining $\Delta_{t+1} = \mathbb{E}[\|\overline{\mathbf{w}}^{(t+1)} - \mathbf{w}^*\|^2]$, $B = 1 + \frac{3\overline{\rho}}{8}$, $C = 32\tau^2 G^2 + \frac{\sigma^2}{m} + 6\overline{\rho}L\Gamma$, $D = 2\Gamma(\widetilde{\rho} - \overline{\rho})$, we have that

$$\Delta_{t+1} \leq (1 - \eta_t\mu B)\Delta_t + \eta_t^2 C + \eta_t D \tag{65}$$

By setting $\Delta_t \leq \frac{\psi}{t+\gamma}$, $\eta_t = \frac{\beta}{t+\gamma}$ and $\beta > \frac{1}{\mu B}$, $\gamma > 0$ by induction we have that

$$\psi = \max\left\{\gamma\|\overline{\mathbf{w}}^{(0)} - \mathbf{w}^*\|^2, \frac{1}{\beta\mu B - 1}\left(\beta^2 C + D\beta(t + \gamma)\right)\right\} \tag{66}$$

Then by the L-smoothness of $F(\cdot)$, we have that

$$\mathbb{E}[F(\overline{\mathbf{w}}^{(t)})] - F^* \leq \frac{L}{2}\Delta_t \leq \frac{L}{2}\frac{\psi}{\gamma + t} \tag{67}$$

## D  PROOF OF THEOREM A.1

With fixed learning rate $\eta_t = \eta$, we can rewrite (65) as

$$\Delta_{t+1} \leq (1 - \eta\mu B)\Delta_t + \eta^2 C + \eta D \tag{68}$$

and with $\eta \leq \min\{\frac{1}{2\mu B}, \frac{1}{4L}\}$ using recursion of (68) we have that

$$\Delta_t \leq (1 - \eta\mu B)^t\Delta_0 + \frac{\eta^2 C + \eta D}{\eta\mu B}(1 - (1 - \eta\mu B)^t) \tag{69}$$

Using $\Delta_t \leq \frac{2}{\mu}(F(\overline{\mathbf{w}}^{(t)}) - F^*)$ and $L$-smoothness, we have that

$$F(\overline{\mathbf{w}}^{(t)}) - F^* \leq \frac{L}{\mu}(1 - \eta\mu B)^t(F(\overline{\mathbf{w}}^{(0)}) - F^*) + \frac{L(\eta C + D)}{2\mu B}(1 - (1 - \eta\mu B)^t) \quad (70)$$

$$= \frac{L}{\mu}\left[1 - \eta\mu\left(1 + \frac{3\overline{\rho}}{8}\right)\right]^t(F(\overline{\mathbf{w}}^{(0)}) - F^*) + \frac{4L(\eta C + D)}{\mu(8 + 3\overline{\rho})}\left[1 - \left[1 - \eta\mu\left(1 + \frac{3\overline{\rho}}{8}\right)\right]^t\right] \quad (71)$$

## E    EXTENSION: GENERALIZATION TO DIFFERENT AVERAGING SCHEMES

While we considered a simple averaging scheme where $\overline{\mathbf{w}}^{(t+1)} = \frac{1}{m}\sum_{k\in\mathcal{S}^{(t)}}\left(\mathbf{w}_k^{(t)} - \eta_t g_k(\mathbf{w}_k^{(t)})\right)$, we can extend the averaging scheme to any scheme $\mathbf{q}$ such that the averaging weights $q_k$ are invariant in time and satisfies $\sum_{k\in\mathcal{S}^{(t)}} q_k = 1$ for any $t$. Note that $\mathbf{q}$ includes the random sampling without replacement scheme introduced by Li et al. (2020) where the clients are sampled uniformly at random without replacement with the averaging coefficients $q_k = p_k K/m$. With such averaging scheme $\mathbf{q}$, we denote the global model for the averaging scheme $q_k$ as $\widehat{\mathbf{w}}^{(t)}$, where $\widehat{\mathbf{w}}^{(t+1)} \triangleq \sum_{k\in\mathcal{S}^{(t)}} q_k\left(\mathbf{w}_k^{(t)} - \eta_t g_k(\mathbf{w}_k^{(t)})\right)$, and the update rule changes to

$$\widehat{\mathbf{w}}^{(t+1)} = \widehat{\mathbf{w}}^{(t)} - \eta_t\widehat{\mathbf{g}}^{(t)} = \widehat{\mathbf{w}}^{(t)} - \eta_t\left(\sum_{k\in\mathcal{S}^{(t)}} q_k g_k(\mathbf{w}_k^{(t)}, \xi_k^{(t)})\right) \quad (72)$$

where $\widehat{\mathbf{g}}^{(t)} = \sum_{k\in\mathcal{S}^{(t)}} q_k g_k(\mathbf{w}_k^{(t)}, \xi_k^{(t)})$. We show that the convergence analysis for the averaging scheme $\mathbf{q}$ is consistent with Theorem 3.1. In the case of the averaging scheme $\mathbf{q}$, we have that Lemma $B.2$ and Lemma $B.3$ shown in Appendix B, each becomes

$$\frac{1}{m}\mathbb{E}[\sum_{k\in\mathcal{S}^{(t)}} \|\widehat{\mathbf{w}}^{(t)} - \mathbf{w}_k^{(t)}\|^2] \leq 16\eta_t^2 m(m-1)\tau^2 G^2 \quad (73)$$

$$\mathbb{E}[\|\widehat{\mathbf{w}}^{(t)} - \mathbf{w}^*\|^2] \leq m\mathbb{E}[\sum_{k\in\mathcal{S}^{(t)}} q_k\|\mathbf{w}_k^{(t)} - \mathbf{w}^*\|^2] \quad (74)$$

Then, using the same method we used for the proof of Theorem 3.1, we have that

$$\mathbb{E}[\|\widehat{\mathbf{w}}^{(t+1)} - \mathbf{w}^*\|^2] \leq \left(1 - \frac{\eta_t\mu}{m}\right)\mathbb{E}[\|\widehat{\mathbf{w}}^{(t)} - \mathbf{w}^*\|^2] + \eta_t^2\sigma^2 m + 16m^2(m-1)\eta_t^2\tau^2 G^2 +$$

$$\underbrace{\mathbb{E}\left[2L\eta_t^2(1+m)\sum_{k\in\mathcal{S}^{(t)}} q_k(F_k(\mathbf{w}_k^{(t)}) - F_k^*) - 2\eta_t\sum_{k\in\mathcal{S}^{(t)}} q_k(F_k(\mathbf{w}_k^{(t)}) - F_k(\mathbf{w}^*))\right]}_{M} \quad (75)$$

By defining the selection skew for averaging scheme $\mathbf{q}$ similar to Definition 5 as

$$\rho_{\mathbf{q}}(\mathcal{S}(\pi, \mathbf{w}), \mathbf{w}') = \frac{\mathbb{E}_{\mathcal{S}(\pi,\mathbf{w})}[\sum_{k\in\mathcal{S}(\pi,\mathbf{w})} q_k(F_k(\mathbf{w}') - F_k^*)]}{F(\mathbf{w}') - \sum_{k=1}^K p_k F_k^*} \geq 0, \quad (76)$$

and

$$\overline{\rho}_{\mathbf{q}} \triangleq \min_{\mathbf{w},\mathbf{w}'} \rho_{\mathbf{q}}(\mathcal{S}(\pi, \mathbf{w}), \mathbf{w}') \quad (77)$$

$$\widetilde{\rho}_{\mathbf{q}} \triangleq \max_{\mathbf{w}} \rho_{\mathbf{q}}(\mathcal{S}(\pi, \mathbf{w}), \mathbf{w}^*) = \frac{\max_{\mathbf{w}} \mathbb{E}_{\mathcal{S}(\pi,\mathbf{w})}[\sum_{k\in\mathcal{S}(\pi,\mathbf{w})} q_k(F_k(\mathbf{w}^*) - F_k^*)]}{\Gamma} \quad (78)$$

With $\eta_t < 1/(2L(1+m))$, using the same methodology for proof of Theorem 3.1 we have that $M$ becomes upper bounded as

$$\mathbb{E}\left[2L\eta_t^2(1+m)\sum_{k\in\mathcal{S}^{(t)}} q_k(F_k(\mathbf{w}_k^{(t)}) - F_k^*) - 2\eta_t\sum_{k\in\mathcal{S}^{(t)}} q_k(F_k(\mathbf{w}_k^{(t)}) - F_k(\mathbf{w}^*))\right] \quad (79)$$

$$\leq -\frac{\eta_t\mu\overline{\rho}_{\mathbf{q}}}{2}\mathbb{E}[\|\widehat{\mathbf{w}}^{(t)} - \mathbf{w}^*\|^2] + 2\eta_t\Gamma(\widetilde{\rho}_{\mathbf{q}} - \overline{\rho}_{\mathbf{q}}) + 16m^2(m-1)\eta_t^2\tau^2 G^2 + 2L\eta_t^2(2+m)\overline{\rho}_{\mathbf{q}}\Gamma \quad (80)$$

Finally we have that

$$
\mathbb{E}[\|\widehat{\mathbf{w}}^{(t+1)} - \mathbf{w}^*\|^2] \leq \left[1 - \eta_t \mu \left(\frac{1}{m} + \frac{\overline{\rho}_{\mathbf{q}}}{2}\right)\right] \mathbb{E}[\|\widehat{\mathbf{w}}^{(t)} - \mathbf{w}^*\|^2] + 2\eta_t \Gamma(\widetilde{\rho}_{\mathbf{q}} - \overline{\rho}_{\mathbf{q}})
$$
$$
+ \eta_t^2 [32m^2(m-1)\tau^2 G^2 + \sigma^2 m + 2L(2+m)\overline{\rho}_{\mathbf{q}}\Gamma]
$$
(81)

By defining $\widehat{\Delta}_{t+1} = \mathbb{E}[\|\widehat{\mathbf{w}}^{(t+1)} - \mathbf{w}^*\|^2]$, $\widehat{B} = \frac{1}{m} + \frac{\overline{\rho}_{\mathbf{q}}}{2}$, $\widehat{C} = 32m^2(m-1)\tau^2 G^2 + \sigma^2 m + 2L(2 + m)\overline{\rho}_{\mathbf{q}}\Gamma$, $\widehat{D} = 2\Gamma(\widetilde{\rho}_{\mathbf{q}} - \overline{\rho}_{\mathbf{q}})$, we have that

$$
\widehat{\Delta}_{t+1} \leq (1 - \eta_t \mu \widehat{B})\widehat{\Delta}_t + \eta_t^2 \widehat{C} + \eta_t \widehat{D}
$$
(82)

Again, by setting $\widehat{\Delta}_t \leq \frac{\psi}{t+\gamma}$, $\eta_t = \frac{\beta}{t+\gamma}$ and $\beta > \frac{1}{\mu \widehat{B}}$, $\gamma > 0$ by induction we have that

$$
\psi = \max\left\{\gamma\|\overline{\mathbf{w}}^{(0)} - \mathbf{w}^*\|^2, \frac{1}{\beta\mu\widehat{B} - 1}\left(\beta^2\widehat{C} + \widehat{D}\beta(t+\gamma)\right)\right\}
$$
(83)

Then by the L-smoothness of $F(\cdot)$, we have that

$$
\mathbb{E}[F(\overline{\mathbf{w}}^{(t)})] - F^* \leq \frac{L}{2}\widehat{\Delta}_t \leq \frac{L}{2}\frac{\psi}{\gamma + t}
$$
(84)

With $\beta = \frac{m}{\mu}$, $\gamma = \frac{4m(1+m)L}{\mu}$ and $\eta_t = \frac{\beta}{t+\gamma}$, we have that

$$
\mathbb{E}[F(\widehat{\mathbf{w}}^{(T)})] - F^* \leq
$$
$$
\underbrace{\frac{1}{(T+\gamma)}\left[\frac{Lm^2(32m(m-1)\tau^2 G^2 + \sigma^2)}{\mu^2\overline{\rho}_{\mathbf{q}}} + \frac{2L^2 m(m+2)\Gamma}{\mu^2} + \frac{L\gamma\|\overline{\mathbf{w}}^{(0)} - \mathbf{w}^*\|^2}{2}\right]}_{\text{Vanishing Error Term}}
$$
$$
+ \underbrace{\frac{2L\Gamma}{\overline{\rho}_{\mathbf{q}}\mu}\left(\frac{\widetilde{\rho}_{\mathbf{q}}}{\overline{\rho}_{\mathbf{q}}} - 1\right)}_{\text{Non-vanishing bias}}
$$
(85)

which is consistent with Theorem 3.1.

## F  EXPERIMENT DETAILS

**Quadratic Model Optimization.** For the quadratic model optimization, we set each local objective function as strongly convex as follows:

$$
F_k(\mathbf{w}) = \frac{1}{2}\mathbf{w}^\top \mathbf{H}_k \mathbf{w} - \mathbf{e}_k^\top \mathbf{w} + \frac{1}{2}\mathbf{e}_k^\top \mathbf{H}_k^{-1}\mathbf{e}_k
$$
(86)

$\mathbf{H}_k \in \mathbb{R}^{v \times v}$ is a diagonal matrix $\mathbf{H}_k = h_k \mathbf{I}$ with $h_k \sim \mathcal{U}(1, 20)$ and $\mathbf{e}_k \in \mathbb{R}^v$ is an arbitrary vector. We set the global objective function as $F(\mathbf{w}) = \sum_{k=1}^{K} p_k F_k(\mathbf{w})$, where the data size $p_k$ follows the power law distribution $P(x; a) = ax^{a-1}$, $0 \leq x \leq 1$, $a = 3$. We can easily show that the optimum for $F_k(\mathbf{w})$ and $F(\mathbf{w})$ is $\mathbf{w}_k^* = \mathbf{H}_k^{-1}\mathbf{e}_k$ and $\mathbf{w}^* = (\sum_{k=1}^{K} p_k \mathbf{H}_k)^{-1}(\sum_{k=1}^{K} p_k \mathbf{e}_k)$ respectively. The gradient descent update rule for the local model of client $k$ in the quadratic model optimization is

$$
\mathbf{w}_k^{(t+1)} = \mathbf{w}_k^{(t)} - \eta(\mathbf{H}_k \mathbf{w}_k^{(t)} - \mathbf{e}_k)
$$
(87)

where the global model is defined as $\overline{\mathbf{w}}^{(t+1)} = \frac{1}{m}\sum_{k \in \mathcal{S}^{(t)}} \mathbf{w}_k^{(t+1)}$. We sample $m = KC$ clients for every round where for each round the clients perform $\tau$ gradient descent local iterations with fixed learning rate $\eta$ and then these local models are averaged to update the global model. For the implementation of $\pi_{\text{adapow-d}}$, $d$ was decreased half from $d = K$ for every 5000 rounds. For all simulations we set $\tau = 2$, $v = 5$, $\eta = 2 \times 10^{-5}$.

For the estimation of $\overline{\rho}$ and $\widetilde{\rho}$ for the quadratic model, we get the estimates of the theoretical $\overline{\rho}$, $\widetilde{\rho}$ values by doing a grid search over a large range of possible $\mathbf{w}, \mathbf{w}'$ for $\rho(\mathcal{S}(\pi, \mathbf{w}), \mathbf{w}')$ and

$\rho(\mathcal{S}(\pi, \mathbf{w}), \mathbf{w}^*)$ respectively. The distribution of $\mathcal{S}(\pi, \mathbf{w})$ is estimated by simulating 10000 iterations of client sampling for each $\pi$ and $\mathbf{w}$.

**Logistic Regression on Synthetic Dataset.** We conduct simulations on synthetic data which allows precise manipulation of heterogeneity. Using the methodology constructed in (Sahu et al., 2019), we use the dataset with large data heterogeneity, Synthetic(1,1). We assume in total 30 devices where the local dataset sizes for each device follows the power law. For the implementation of $\pi_{\text{adapow-d}}$, $d$ was decreased to $d = m$ from $d = K$ at half the entire communication rounds. We set the mini batch-size to 50 with $\tau = 30$, and $\eta = 0.05$, where $\eta$ is decayed to $\eta/2$ every 300 and 600 rounds.

**DNN on FMNIST Dataset.** We train a deep multi-layer perceptron network with two hidden layers on the FMNIST dataset (Xiao et al., 2017). We construct the heterogeneous data partition amongst clients using the Dirichlet distribution $\text{Dir}_K(\alpha)$ (Hsu et al., 2019), where $\alpha$ determines the degree of the data heterogeneity across clients (the data size imbalance and degree of label skew across clients). Smaller $\alpha$ indicates larger data heterogeneity. For all experiments we use mini-batch size of $b = 64$, with $\tau = 30$ and $\eta = 0.005$, where $\eta$ is decayed by half for every 150, 300 rounds. We experiment with three different seeds for the randomness in the dataset partition across clients and present the averaged results.

All experiments are conducted with clusters equipped with one NVIDIA TitanX GPU. The number of clusters we use vary by $C$, the fraction of clients we select. The machines communicate amongst each other through Ethernet to transfer the model parameters and information necessary for client selection. Each machine is regarded as one client in the federated learning setting. The algorithms are implemented by PyTorch.

**Pseudo-code of the variants of pow-d: cpow-d and rpow-d.** We here present the pseudo-code for $\pi_{\text{cpow-d}}$ and $\pi_{\text{rpow-d}}$. Note that the pseudo-code for $\pi_{\text{cpow-d}}$ in Algorithm 1 can be generalized to the algorithm for $\pi_{\text{pow-d}}$, by changing $\frac{1}{|\widehat{\xi}_k|} \sum_{\xi \in \widehat{\xi}_k} f(\mathbf{w}, \xi)$ to $F_k(\mathbf{w})$.

---

**Algorithm 1** Pseudo code for cpow-d: computation efficient variant of pow-d

1: **Input**: $m$, $d$, $p_k$ for $k \in [K]$, mini-batch size $b = |\widehat{\xi}_k|$ for computing $\frac{1}{|\widehat{\xi}_k|} \sum_{\xi \in \widehat{\xi}_k} f(\mathbf{w}, \xi)$

2: **Output**: $\mathcal{S}^{(t)}$

3: **Initialize**: empty sets $\mathcal{S}^{(t)}$ and $\mathcal{A}$

4: **Global server do**

5:    Get $\mathcal{A} = \{d$ indices sampled without replacement from $[K]$ by $p_k\}$

6:    Send the global model $\overline{\mathbf{w}}^{(t)}$ to the $d$ clients in $\mathcal{A}$

7:    Receive $\frac{1}{|\widehat{\xi}_k|} \sum_{\xi \in \widehat{\xi}_k} f(\mathbf{w}, \xi)$ from all clients in $\mathcal{A}$

8:    Get $\mathcal{S}^{(t)} = \{m$ clients with largest $\frac{1}{|\widehat{\xi}_k|} \sum_{\xi \in \widehat{\xi}_k} f(\mathbf{w}, \xi)$ (break ties randomly)$\}$

9: **Clients in $\mathcal{A}$ in parallel do**

10:    Create mini-batch $\widehat{\xi}_k$ from sampling $b$ samples uniformly at random from $\mathcal{B}_k$ and compute $\frac{1}{|\widehat{\xi}_k|} \sum_{\xi \in \widehat{\xi}_k} f(\mathbf{w}, \xi)$ and send it to the server

11: **return** $\mathcal{S}^{(t)}$

---

---

**Algorithm 2** Pseudo code for `rpow-d`: computation & communication efficient variant of `pow-d`

1: **Input**: $m$, $d$, $p_k$ for $k \in [K]$
2: **Output**: $\mathcal{S}^{(t)}$
3: **Initialize**: empty sets $\mathcal{S}^{(t)}$ and $\mathcal{A}$, and list $A_{\text{tmp}}$ with $K$ elements all equal to `inf`
4: **All client $k \in \mathcal{S}^{(t-1)}$ do**
5:     For $t \bmod \tau = 0$, send $\frac{1}{\tau b} \sum_{l=t-\tau+1}^{t} \sum_{\xi \in \xi_k^{(l)}} f(\mathbf{w}_k^{(l)}, \xi)$ to the server with its local model

6: **Global server do**
7:     Receive and update $A_{\text{tmp}}[k] = \frac{1}{\tau b} \sum_{l=t-\tau+1}^{t} \sum_{\xi \in \xi_k^{(l)}} f(\mathbf{w}_k^{(l)}, \xi)$ for $k \in \mathcal{S}^{(t-1)}$
8:     Get $\mathcal{A} = \{d$ indices sampled without replacement from $[K]$ by $p_k\}$
9:     Get $\mathcal{S}^{(t)} = \{m$ clients with largest values in $[A_{\text{tmp}}[i]$ for $i \in \mathcal{A}]$, (break ties randomly)$\}$
10: **return** $\mathcal{S}^{(t)}$

---

# G    ADDITIONAL EXPERIMENT RESULTS

## G.1    SELECTED CLIENT PROFILE

We further visualize the difference between our proposed sampling strategy $\pi_{\text{pow-d}}$ and the baseline scheme $\pi_{\text{rand}}$ by showing the selected frequency ratio of the clients for $K = 30$, $C = 0.1$ for the quadratic simulations in Figure 7. Note that the selected ratio for $\pi_{\text{rand}}$ reflects each client's dataset size. We show that the selected frequencies of clients for $\pi_{\text{pow-d}}$ are not proportional to the data size of the clients, and we are selecting clients frequently even when they have relatively low data size like client 6 or 22. We are also not necessarily frequently selecting the clients that have the highest data size such as client 26. This aligns well with our main motivation of POWER-OF-CHOICE that weighting the clients' importance based on their data size does not achieve the best performance, and rather considering their local loss values along with the data size better represents their importance. Note that the selected frequency for $\pi_{\text{rand}}$ is less biased than $\pi_{\text{pow-d}}$.

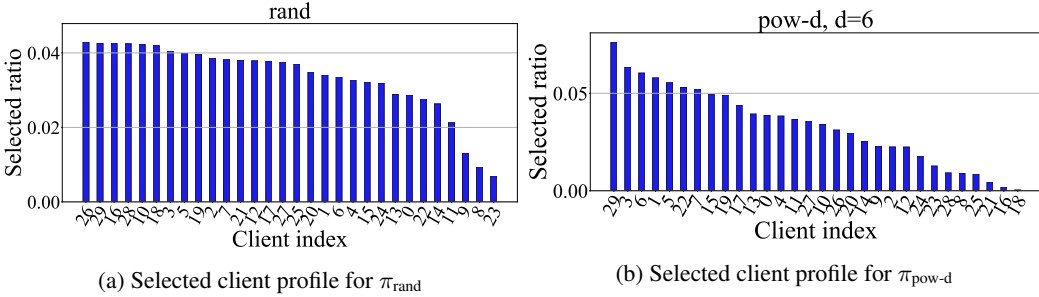

(a) Selected client profile for $\pi_{\text{rand}}$          (b) Selected client profile for $\pi_{\text{pow-d}}$

Figure 7: Clients' selected frequency ratio for optimizing the quadratic model for $\pi_{\text{rand}}$ and $\pi_{\text{pow-d}}$ with $K = 30$, $C = 0.1$. The selected ratio is sorted in the descending order.

## G.2    COMMUNICATION AND COMPUTATION EFFICIENCY WITH LARGER DATA HETEROGENEITY

In Table 2, we show the communication and computation efficiency of POWER-OF-CHOICE for $\alpha = 2$, as we showed for $\alpha = 0.3$ in Table 1 in Section 5. With $C = 0.03$ fraction of clients, $\pi_{\text{pow-d}}$, $\pi_{\text{cpow-d}}$, and $\pi_{\text{rpow-d}}$ have better test accuracy of at least approximately 10% higher test accuracy performance than $(\pi_{\text{rand}}, C = 0.1)$. $R_{60}$ for $\pi_{\text{pow-d}}$, $\pi_{\text{cpow-d}}$, $\pi_{\text{rpow-d}}$ is 0.61, 0.66, 0.73 times that of $(\pi_{\text{rand}}, C = 0.1)$ respectively. This indicates that we can reduce the number of communication rounds by at least 0.6 using 1/3 of clients compared to $(\pi_{\text{rand}}, C = 0.1)$ and still get higher test accuracy performance. The computation time $t_{\text{comp}}$ for $\pi_{\text{cpow-d}}$ and $\pi_{\text{rpow-d}}$ with $C = 0.03$ is smaller than that of $(\pi_{\text{rand}}, C = 0.1)$.

Table 2: Comparison of $R_{60}$, $t_{\text{comp}}$ (sec), and test accuracy (%) for different sampling strategies with $\alpha = 2$. The ratio $R_{60}$ / ($R_{60}$ for rand, $C = 0.1$) and $t_{\text{comp}}$ / ($t_{\text{comp}}$ for rand, $C = 0.1$) are each shown in the parenthesis.

| | $C = 0.1$ | $C = 0.03$ | | | | |
|---|---|---|---|---|---|---|
| | rand | rand | pow-d, $d = 6$ | cpow-d, $d = 6$ | rpow-d, $d = 50$ | afl |
| $R_{60}$ | 135 | 136(1.01) | **82 (0.61)** | **89 (0.66)** | 99(0.73) | 131(0.97) |
| $t_{\text{comp}}$ | 0.42 | 0.36(0.85) | **0.46 (1.08)** | **0.38 (0.88)** | 0.36(0.86) | 0.36(085) |
| Test Acc. | **63.50±2.74** | 66.03±1.47 | **73.81±1.14** | **73.36±1.17** | 72.52±0.89 | 70.64±1.99 |

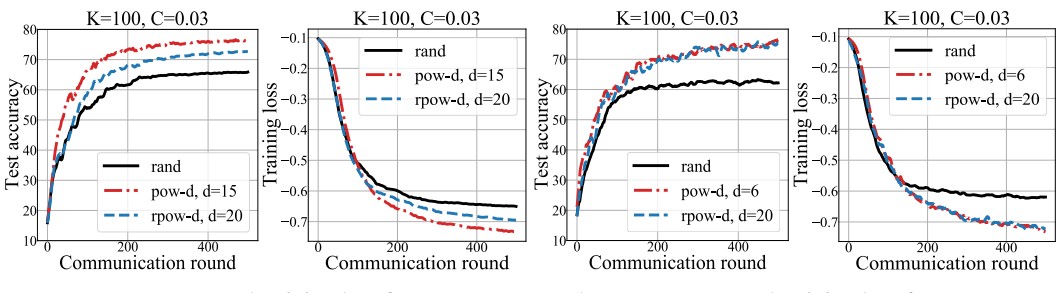

(a) Test accuracy and training loss for $\alpha = 2$      (b) Test accuracy and training loss for $\alpha = 0.3$

Figure 8: Test accuracy and training loss in the virtual environment where clients have intermittent availability for $K = 100$, $C = 0.03$ with $\pi_{\text{rand}}$, $\pi_{\text{pow-d}}$, and $\pi_{\text{rpow-d}}$ on the FMNIST dataset. For both $\alpha = 2$ and $\alpha = 3$, $\pi_{\text{pow-d}}$ achieves approximately 10% higher test accuracy than $\pi_{\text{rand}}$.

## G.3    INTERMITTENT CLIENT AVAILABILITY

In real world scenarios, certain clients may not be available due to varying availability of resources such as battery power or wireless connectivity. Hence we experiment with a virtual scenario, where amongst $K$ clients, for each communication round, we select clients alternately from one group out of two fixed groups, where each group has $0.5K$ clients. This altering selection reflects a more realistic client selection scenario where, for example, we have different time zones across clients. For each communication round, we select 0.1 portion of clients from the corresponding group uniformly at random and exclude them from the client selection process. This random exclusion of certain clients represents the randomness in the client availability within that group for cases such as low battery power or wireless connectivity. In Figure 8 we show that $\pi_{\text{pow-d}}$ and $\pi_{\text{rpow-d}}$ achieves 10% and 5% test accuracy improvement respectively compared to $\pi_{\text{rand}}$ for $\alpha = 2$. For $\alpha = 3$, both $\pi_{\text{pow-d}}$ and $\pi_{\text{rpow-d}}$ shows 10% improvement. Therefore, we demonstrate that POWER-OF-CHOICE also performs well in a realistic scenario where clients are available intermittently.

## G.4    RESULTS FOR DNN ON NON-IID PARTITIONED EMNIST DATASET

To provide further validation of the consistency in our results of $\pi_{\text{pow-d}}$ and its variants on the FMNIST dataset, we present additional experiment results on the EMNIST dataset sorted by digits with $K = 500$, $C = 0.03$. We train a deep multi-layer perceptron network with two hidden layers on the dataset partitioned heterogeneously across the clients in the same way as for the FMNIST dataset. For all experiments, we use $b = 64$, $\tau = 30$, and $\eta = 0.005$ where $\eta$ is decayed by half at round 300.

In Figure 9, we show that $\pi_{\text{pow-d}}$ performs with significantly higher test accuracy than $\pi_{\text{rand}}$ for varying $d$ for both $\alpha = 2$ and 0.3. For $\alpha = 2$, $\pi_{\text{afl}}$ is able to follow the performance of $\pi_{\text{pow-d}}$ in the later communication rounds, but is slower in achieving the same test accuracy than $\pi_{\text{pow-d}}$. Moreover, in Figure 10, we show that $\pi_{\text{cpow-d}}$ works as good as $\pi_{\text{pow-d}}$ for both large and small data heterogeneity. The performance of $\pi_{\text{rpow-d}}$ falls behind $\pi_{\text{pow-d}}$ and $\pi_{\text{cpow-d}}$ for smaller data heterogeneity, whereas for larger data heterogeneity, $\pi_{\text{rpow-d}}$ is able to perform similarly with $\pi_{\text{pow-d}}$ and $\pi_{\text{cpow-d}}$.

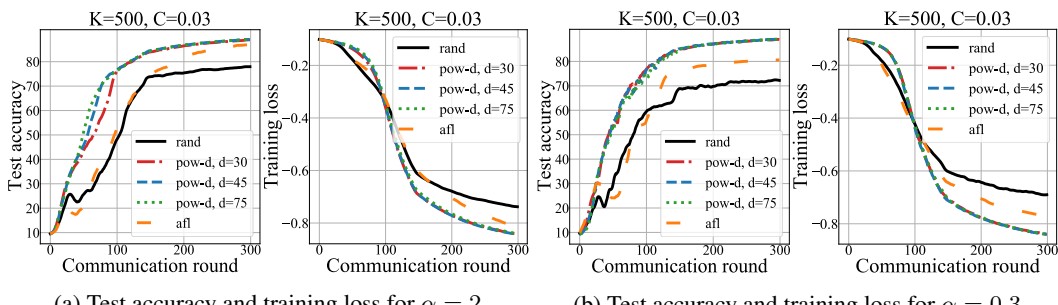

Figure 9: Test accuracy and training loss for different sampling strategies for $K = 500$, $C = 0.03$ with $\pi_{\text{rand}}$, $\pi_{\text{pow-d}}$, and $\pi_{\text{afl}}$ on the EMNIST dataset.

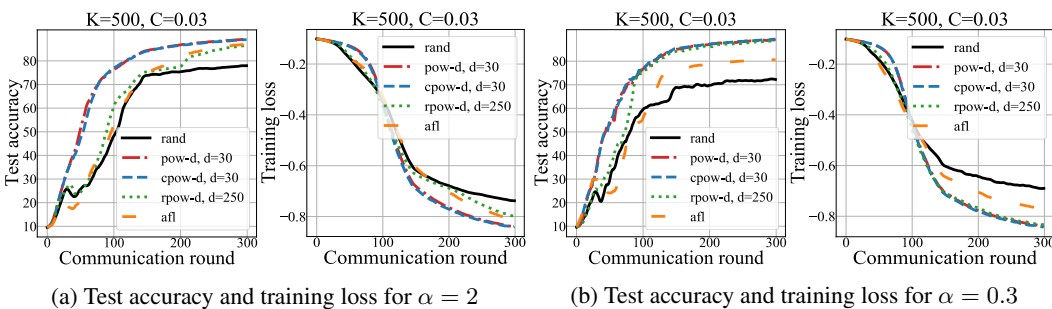

Figure 10: Test accuracy and training loss for different sampling strategies for $K = 500$, $C = 0.03$ with $\pi_{\text{rand}}$, $\pi_{\text{pow-d}}$, $\pi_{\text{cpow-d}}$, $\pi_{\text{rpow-d}}$, and $\pi_{\text{afl}}$ on the EMNIST dataset.

### G.5 EFFECT OF THE FRACTION OF SELECTED CLIENTS

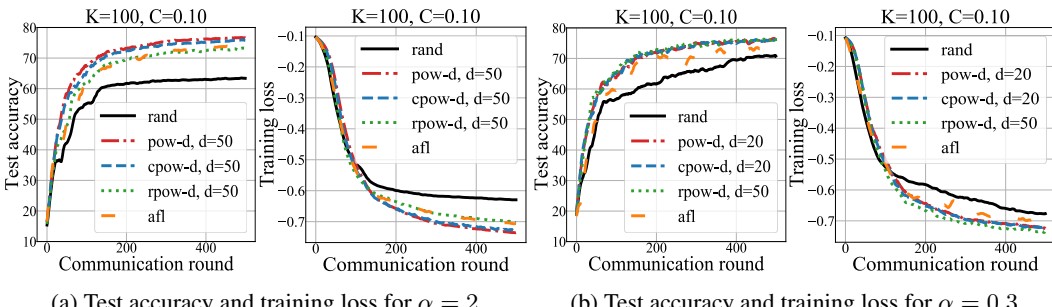

Figure 11: Test accuracy and training loss for different sampling strategies for $K = 100$, $C = 0.1$ with $\pi_{\text{rand}}$, $\pi_{\text{pow-d}}$, $\pi_{\text{cpow-d}}$, $\pi_{\text{rpow-d}}$, and $\pi_{\text{afl}}$ on the FMNIST dataset. For larger $C = 0.1$, $\pi_{\text{pow-d}}$ performs with 15% and 5% higher test accuracy than $\pi_{\text{rand}}$ for $\alpha = 2$ and $\alpha = 0.3$ respectively.

In Figure 11, for larger $C = 0.1$ with $\alpha = 2$, the test accuracy improvement for $\pi_{\text{pow-d}}$ is even higher than the case of $C = 0.03$ with approximately 15% improvement. $\pi_{\text{cpow-d}}$ performs slightly lower in test accuracy than $\pi_{\text{pow-d}}$ but still performs better than $\pi_{\text{rand}}$ and $\pi_{\text{afl}}$. $\pi_{\text{rpow-d}}$ performs as well as $\pi_{\text{afl}}$. For $\alpha = 0.3$, $\pi_{\text{pow-d}}$, $\pi_{\text{cpow-d}}$, and $\pi_{\text{rpow-d}}$ have approximately equal test accuracy performance, higher than $\pi_{\text{rand}}$ by 5%. The POWER-OF-CHOICE strategies all perform slightly better than $\pi_{\text{afl}}$. Therefore we show that POWER-OF-CHOICE performs well for selecting a larger fraction of clients, i.e., when we have larger $C = 0.1 > 0.03$.

### G.6 EFFECT OF THE LOCAL EPOCHS AND MINI-BATCH SIZE

We present the experiment results elaborated in Section 5 for the different hyper-parameter settings $(b, \tau) \in \{(128, 30), (64, 100)\}$ in Figure 12, 13, 14, and 15 below.

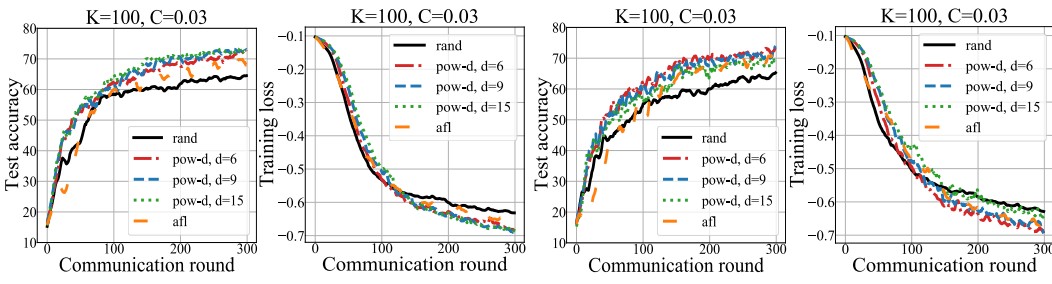

(a) Test accuracy and training loss for $\alpha = 2$        (b) Test accuracy and training loss for $\alpha = 0.3$

Figure 12: Test accuracy and training loss for $\pi_{\text{rand}}$, $\pi_{\text{pow-d}}$, and $\pi_{\text{afl}}$ for $K = 100$, $C = 0.03$ on the FMNIST dataset with mini-batch size $b = 128$ and $\tau = 30$.

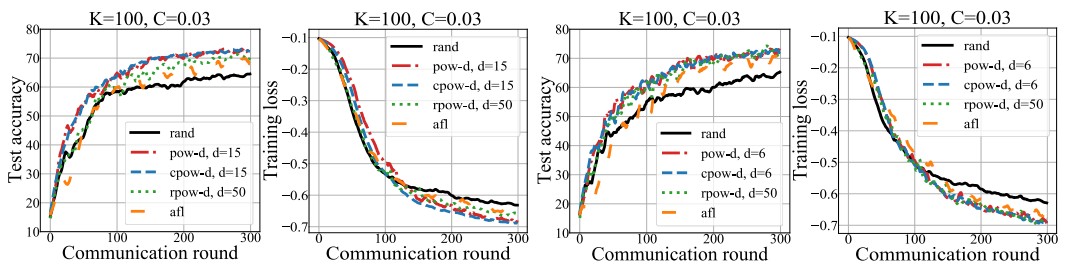

(a) Test accuracy and training loss for $\alpha = 2$        (b) Test accuracy and training loss for $\alpha = 0.3$

Figure 13: Test accuracy and training loss for $\pi_{\text{rand}}$, $\pi_{\text{pow-d}}$, $\pi_{\text{cpow-d}}$, $\pi_{\text{rpow-d}}$, and $\pi_{\text{afl}}$ for $K = 100$, $C = 0.03$ on the FMNIST dataset with mini-batch size $b = 128$ and $\tau = 30$.

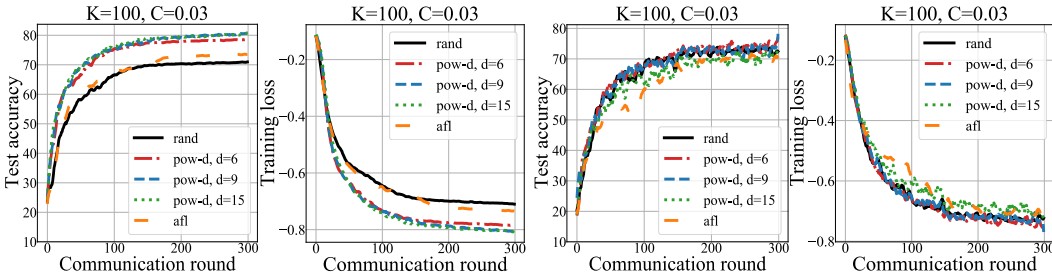

(a) Test accuracy and training loss for $\alpha = 2$        (b) Test accuracy and training loss for $\alpha = 0.3$

Figure 14: Test accuracy and training loss for $\pi_{\text{rand}}$, $\pi_{\text{pow-d}}$, and $\pi_{\text{afl}}$ for $K = 100$, $C = 0.03$ on the FMNIST dataset with mini-batch size $b = 64$ and $\tau = 100$.

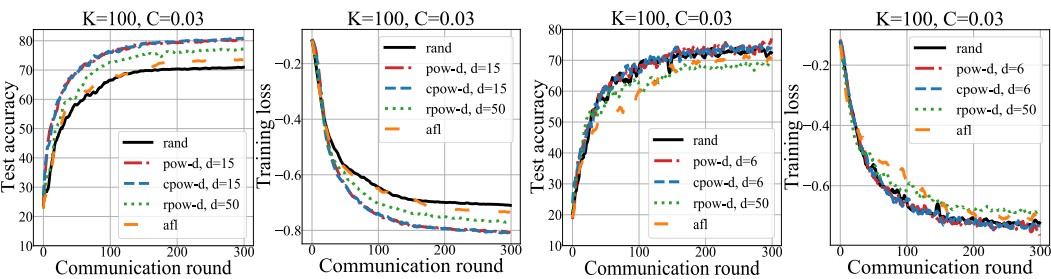

(a) Test accuracy and training loss for $\alpha = 2$        (b) Test accuracy and training loss for $\alpha = 0.3$

Figure 15: Test accuracy and training loss for $\pi_{\text{rand}}$, $\pi_{\text{pow-d}}$, $\pi_{\text{cpow-d}}$, $\pi_{\text{rpow-d}}$, and $\pi_{\text{afl}}$ for $K = 100$, $C = 0.03$ on the FMNIST dataset with mini-batch size $b = 64$ and $\tau = 100$.

