# OpenReview forum: "Client Selection in Federated Learning: Convergence Analysis and Power-of-Choice Selection Strategies"
_ICLR.cc/2021/Conference — Reject_

### Official Review · AnonReviewer4 · 2020-10-25
**An interesting submission with limited theoretical novelty**

**Rating:** 4
**Confidence:** 4

**Review:**

This work investigates federated optimization considering data heterogeneity, communication and computation limitations, and partial client participation. In contrast to past works, this paper focuses on deeper understanding of the effect of partial client participation on the convergence rate by considering biased client participation. The paper provides convergence analysis for any biased selection strategy, showing that the rate is composed of vanishing error term and non-vanishing bias term. The obtained rates explicitly show the effect of client selection strategy and the trade-off between convergence speed and the solution bias. Then it proposes a parametric family of biased selection strategy, called power-of-choice, which aims to speed up the convergence of the error term at the cost of possibly bigger bias term. Experiments are provided to highlight the benefits of the proposed pow-d strategy over the standard unbiased selection strategies.

In terms of the assumptions on the loss functions, there is no improvement as the same assumptions are used in recent works [4,5]. The Assumption 3.4 seems problematic as together with Assumption 3.3 it implies that gradients of local loss functions $F_k$ are uniformly bounded. This is in conflict with Assumption 3.2 as local losses $F_k$ are assumed to be strongly convex. For instance, in papers [1,2,3] the Assumption 3.4 is not needed. Can the current theory be extended by relaxing Assumption 3.4 ?

For Definition 3.1 on local-global objectivity gap, it should be mentioned that this notion was defined and used earlier in [4]. It should be credited properly.

The metrics $\bar{\rho}$ and $\tilde{\rho}$ describing the skewness of client selection strategy, and their explicit effect on convergence rate (7) are interesting. It is also nice that the theory, e.g. Theorem 3.1, recovers the result of unbiased client selection case without any solution bias. Since Theorem 3.1 is generic and works for any selection strategy, it does not explicitly show a clear benefit of biased selection strategy over unbiased one. Biased selection strategy reduces the vanishing error term by $\bar{\rho}$, and adds a non-vanishing bias $Q(\bar{\rho}, \tilde{\rho})=O(\frac{\tilde{\rho}}{\bar{\rho}}-1)$. So, based on the rate given in Theorem 3.1, biased selection is beneficial only if $\bar{\rho}>1$ and the difference $\tilde{\rho}-\bar{\rho}$ is small (formally it should be $O((\bar{\rho}-1)/T))$. The question is, can such biased selection strategy be designed so that it is theoretically better (or at least not worse) than unbiased selection strategy ?

The proof of the main Theorem 3.1 largely follows the proof of Theorem 1 of [4]. It seems that the proof of Theorem 3.1 deviates from the proof of Theorem 1[4] only in derivations (53)-(63), where biased-ness of the client selection kicks in and skewness metrics get involved. Such overlaps in proof techniques should be mentioned and some discussion is needed to highlight the novelty of the proposed analysis.

Using insights from Theorem 3.1, the paper proposes a client selection strategy (with two practical variations), called power-of-choice, which aims to speed up the convergence of vanishing error by maximizing rho_bar. However, it is not clear how the other metric $\tilde{\rho}$ would behave in this selection strategy. Although inspired from Theorem 3.1, I view the power-of-choice selection strategy as a heuristic idea as (i) it focuses only on the vanishing error term and does nothing to minimize the bias term, (ii) no theoretical estimates are developed for the metrics $\bar{\rho}, \tilde{\rho}$ in this specific selection strategy.

With points made above, the theoretical contribution of the paper is weak. I think, the paper would largely improved if it were developed a way on how to avoid the bias term while using biased selection. One possible way is to design a suitable selection strategy which allows to bound the bias. For instance, from the experiment shown on Figure 2.b, it is tempting to show that for the proposed pow-d selection strategy the bias term $(\frac{\tilde{\rho}}{\bar{\rho}} - 1)$ is $O(d/K)$, which can be controlled via parameter $d$. Another possible way might be to incorporate some mechanism on top of FedAvg algorithm similar to what error compensation(or error feedback) does for biased (contractive) compression operators.

Experiments are okay, but they use up to 100 clients which is far from the scale of typical FL applications. In addition, in the experiment shown in Figure 4.b, the training loss seems to be decreasing steadily and non vanishing bias term is not dominated there. As the vanishing error term reduces by choosing larger $d$, why in this experiment smaller $d=6$ performs better than larger $d=15$ ?


[1] A Khaled, K Mishchenko, and P Richtárik. Tighter theory for local SGD on identical and heterogeneous data. In The 23rd International Conference on Artificial Intelligence and Statistics (AISTATS 2020), 2020.

[2] Blake Woodworth, Kumar Kshitij Patel, Sebastian U Stich, Zhen Dai, Brian Bullins, H Brendan McMahan, Ohad Shamir, and Nathan Srebro. Is local SGD better than minibatch SGD? arXiv preprint arXiv:2002.07839, 2020.

[3] Anastasia Koloskova, Nicolas Loizou, Sadra Boreiri, Martin Jaggi, and Sebastian U Stich. A unified theory of decentralized SGD with changing topology and local updates. arXiv preprint arXiv:2003.10422, 2020.

[4] Xiang Li, Kaixuan Huang, Wenhao Yang, Shusen Wang, and Zhihua Zhang. On the convergence of fedavg on non-iid data. In International Conference on Learning Representations (ICLR), July 2020.

[5] Yichen Ruan, Xiaoxi Zhang, Shu-Che Liang, and Carlee Joe-Wong. Towards flexible device participation in federated learning for non-iid data. ArXiv, 2020.

---

> ### Author Response · Authors · 2020-11-18
> **Response to reviewer 4 (reviewer's responses are in italics)**
>
> We thank the reviewer for the positive feedback and detailed constructive suggestions on our paper. We believe that we have addressed all the reviewer’s comments fully, and hope the reviewer can increase the score in reflect of our responses and improvements in the paper. We address the reviewer's feedback accordingly as follows:
>
> Q1: *Assumption 3.4 seems problematic as together with Assumption 3.3 it implies that gradients of local loss functions Fk are uniformly bounded. This is in conflict with Assumption 3.2 as local losses Fk are assumed to be strongly convex. For instance, in papers [1,2,3] the Assumption 3.4 is not needed. Can the current theory be extended by relaxing Assumption 3.4 ?*
>
> A1:
> - We acknowledge that our analysis is based on assumption 3.4, uniformly bounded stochastic gradients. We would like to stress that the problem of the convergence of biased client selection strategies is non-trivial and the papers the reviewer mentioned only consider unbiased client selection strategies. Although we started off with the corresponding assumptions, we intend to relax these assumptions in our future work (i.e., non-convex scenarios without assumption 3.4). We also would like to stress that our paper's significance is in the novelty of the first analysis for biased client selection strategies that have not been looked into before and the insight it gives to the FL community along with the effectiveness of our proposed power-of-d client strategy. To the best of our knowledge, there has been no work giving convergence insights for biased client selection strategies that are cognizant of the training process for either strongly-convex or non-convex functions. Our work presents the first insight in this area, and we intend to extend the analysis to non-convex scenarios as the next step. We do not claim novelty in the analysis technique itself.
>
> Q2: *For Definition 3.1 on local-global objectivity gap, it should be mentioned that this notion was defined and used earlier in [4]. It should be credited properly.*
>
> A2:
> - We thank the reviewer for pointing this out! We completely agree that the definition of the local-global objectivity gap should be duly credited to [4]. We have included a sentence about this in the updated version, which will be uploaded soon.
>
> Q3: *The question is, can such biased selection strategy be designed so that it is theoretically better (or at least not worse) than unbiased selection strategy?*
>
> A3:
> - Similar point with Q5 below, addressed in reply for Q5.
>
> Q4: *Such overlaps in proof techniques should be mentioned and some discussion is needed to highlight the novelty of the proposed analysis.*
>
> A4:
> - Thank you very much for pointing this out! We completely agree with the reviewer that the overlap with proof techniques used in [4] should be duly acknowledged. We will certainly discuss this in the updated version and will highlight that the main novelty of our paper as compared to [4] is that we consider biased selection policies and show how the selection skew affects convergence.
>
> Q5: *However, it is not clear how the other metric ρ~ would behave in this selection strategy. Although inspired by Theorem 3.1, I view the power-of-choice selection strategy as a heuristic idea as (i) it focuses only on the vanishing error term and does nothing to minimize the bias term*
>
> A5:
> - We appreciate the reviewer's opinion regarding the bias term and the suggestions for improving the pow-d strategy. We would first like to highlight that the pow-d strategy spans a natural trade-off between convergence speed and selection bias. The choice of $d$ controls this trade-off. For example, $d=m$ can entirely eliminate the selection bias with seeing no benefit in convergence speed. Increasing $d$ will improve the convergence speed but may increase the non-vanishing error term. In our view, the understanding of this trade-off is one of the main contributions of our paper.
>
>   Nevertheless, we agree with the reviewer's point that the paper can be improved with a selection strategy that tackles the selection bias while gaining the benefit of convergence speed. This can be simply achieved by adaptively reducing $d$ of our pow-d strategy during the course of training. Henceforth, we included the proposition of 'adapow-d' in the updated paper, which decreases $d$ throughout training eventually dropping down to $d=m$. This enables the training loss to eventually converge to the minimal point without the selection bias, while also enjoying the convergence speed in the initial training phase with larger $d>m$. We verify the performance of 'adapow-d' in both the quadratic and synthetic simulations wherein both simulations the training loss converges without the selection bias while having faster convergence speed than the baseline strategy. We plan to add these result plots of ‘adapow-d’ to the updated version of the paper.

---

> > ### Author Response · Authors · 2020-11-18
> > **(cont'd) Response to reviewer 4 (reviewer's responses are in italics)**
> >
> > Q6: *(ii) no theoretical estimates are developed for the metrics ρ¯,ρ~ in this specific selection strategy.*
> >
> > A6:
> > - We do provide the theoretical estimates of $\overline{\rho}$ and $\widetilde{\rho}$ in Fig 2(b) for the proposed pow-d and baseline selection strategy for varying $d$ and $K$. Perhaps the reviewer missed seeing it -- we will make it more prominent in an updated version. Our theory matches well with the quadratic simulations in that as $d$ increases the estimated $\overline{\rho}$ increases but the selection bias $\widetilde{\rho}/\overline{\rho}$ also increases. The methodology of getting these estimated theoretical values is elaborated in Appendix F.
> >
> > Q7: *I think, the paper would largely improved if it were developed a way on how to avoid the bias term while using biased selection.*
> >
> > A7:
> > - While our main contribution is the theoretical insight of biased client selection in FL and the proposed selection strategy pow-d that utilizes this insight, as elaborated in A5 we included the version of pow-d ('adapow-d') to be extended to also be able to eventually eliminate selection bias. Accordingly, we would very much appreciate it if the reviewer could re-evaluate the review score considering our paper's contribution and novelty in being the first paper to analyze biased client selection strategies in FL and proposing an effective selection strategy to enjoy the properties of biased client selection.
> >
> > Q8: *Experiments are okay, but they use up to 100 clients which is far from the scale of typical FL applications.*
> >
> > A8:
> > - To verify pow-d performance in the more realistic FL scenarios with the number of clients $K>100$, we add a benchmark experiment with $K=500, C=0.03$ with the EMNIST dataset (‘by digits’), and will add their results to our updated version of the paper. Also, note that in order to emulate a realistic FL scenario with large $K$, we intentionally used a small fraction C=0.03 in our experiments and showed that the proposed client selection strategies work well.
> >
> > Q9: *In addition, in the experiment shown in Figure 4.b, the training loss seems to be decreasing steadily and non vanishing bias term is not dominated there. As the vanishing error term reduces by choosing larger d, why in this experiment smaller d=6 performs better than larger d=15?*
> >
> > A9:
> > - To clarify and elaborate on Figure 4 as a whole, with larger data heterogeneity, intuitively the selection bias does not reduce by larger d, but as a matter of fact increases because we are putting more bias on certain clients which are most likely to have statistically different data than other clients. This intuition is consistent with the test and training results in Fig4b. However, in cases of less data heterogeneity as in Fig4a., larger d can work better since the statistical discrepancy amongst the datasets of the clients are small, and selecting larger loss clients does not lead to a particular bias towards that client and generalizes well.

---

> > > ### Comment · AnonReviewer4 · 2020-11-19
> > > **Thanks the author(s) for the response.**
> > >
> > > A1: Papers [1,2,3] were mentioned to highlight that the conflict of assumptions used in the paper might be avoided. I got confused with the statements: on the one hand (our paper's significance is in the novelty of the first analysis for biased client selection strategies that have not been looked into before) and on the other hand (We do not claim novelty in the analysis technique itself.)
> > >
> > > A6: By theoretical estimates I meant lower/upper bounds that can be formally proved. Fig 2(b) and figures in general cannot provide theoretical  estimates. In Fig 2(b) theoretical values are approximated in the simple case of quadratic optimization.
> > >
> > > A9: Ok, larger d increases selection bias and non-vanishing error term (also heterogeneity favors to stronger increase). But as I mentioned in the question non-vanishing bias term is not dominated in Fig4(b) and is not explaining the situation.
> > >
> > > One more point related to theoretical novelty. In Theorem 6 of [arXiv:2006.11077] authors consider partial participation with any (fixed) distribution over nodes (section 4.2), where, in contrast to the current submission, communication is done in every iteration but in compressed form. Note that in that Theorem 6, (1) instead of uniformly bounded second moment assumption they use more relaxed one (assumption 3), (2) partial participation can potentially be biased and (3) more importantly, the rate is O(1/T) without any non-vanishing bias term. It seems that the non-vanishing bias term is eliminated by adjusting the aggregation step at the server. With this point, the presence of non-vanishing bias term in Theorem 3.1 is not justified by the usage of biased selection strategy as it seems to be a defect of analysis/method.

---

> > > > ### Author Response · Authors · 2020-11-20
> > > > **Response to reviewer 4 (reviewer's responses are in italics)**
> > > >
> > > > Reviewer's first response to A1: *Papers [1,2,3] were mentioned to highlight that the conflict of assumptions used in the paper might be avoided. I got confused with the statements: on the one hand (our paper's significance is in the novelty of the first analysis for biased client selection strategies that have not been looked into before) and on the other hand (We do not claim novelty in the analysis technique itself.)*
> > > >
> > > > AA1:
> > > > - Sorry about the confusion. We clarify that we do not claim introducing an entirely new analysis technique (since our first part of the derivation follows techniques used in https://arxiv.org/abs/1907.02189) or relaxing previous assumptions. The main novelty lies in introducing the concept of selection skew that has not been seen before and utilizing it in the convergence analysis to analyze the effect of biased client selection strategies on the convergence of federated learning. We show that judicious use of selection skew (as demonstrated by the proposed pow-d strategy) can improve convergence speed. While relaxing the assumptions will improve the theoretical guarantees, we believe that it won’t change the key message of this paper.
> > > >
> > > > Reviewer's first response to A6: *By theoretical estimates I meant lower/upper bounds that can be formally proved. Fig 2(b) and figures in general cannot provide theoretical estimates. In Fig 2(b) theoretical values are approximated in the simple case of quadratic optimization.*
> > > >
> > > > AA6:
> > > > - Thanks for clarifying your question! We believe that finding lower/upper bounds on the selection skew, while interesting, is beyond the scope of this paper. We stress that our analysis encompasses a general class of client selection strategies that follows the training progress which makes it a hard problem to present concrete generalized bounds on the rho values. We believe that the estimates are shown in Fig2(b) are sufficient for the scope of this paper.
> > > >
> > > > Reviewer's first response to A9: *Ok, larger d increases selection bias and non-vanishing error term (also heterogeneity favors stronger increase). But as I mentioned in the question, the non-vanishing bias term is not dominated in Fig4(b) and is not explaining the situation.*
> > > >
> > > > AA9:
> > > > We are a bit confused of what the reviewer means by “the non-vanishing bias term is not dominated in Fig4(b)”, but from our understanding we answer as follows:
> > > > - In Fig4(b), as $d$ increases, the non-vanishing bias term (selection bias) gets more dominant than when $d$ is small, which shows there is the presence of selection bias. The reason that it may not be as dominant as expected is because the bias changes throughout communication rounds: we are not forcing the bias to be towards a same single client or a same single subset of clients for all communication rounds. We are selecting different largest loss clients within the different set of $d$ clients for a different global model every communication round, and therefore we are changing which clients we are biasing towards, every round. This may result in the non-vanishing bias term not being dominant as expected.
> > > >
> > > > - Linking back to Theorem 3.1 presented in our paper, due to the largely non-convex landscape of the DNN experiments, the theorem does not directly apply to the DNN experiments. However, we performed the experiment as a sanity check of the effectiveness of our proposed pow-d in the non-convex problems.
> > > >
> > > > Please let us know if we have misunderstood the reviewer’s question!

---

> > > > > ### Author Response · Authors · 2020-11-20
> > > > > **(cont'd) Response to reviewer 4 (reviewer's responses are in italics)**
> > > > >
> > > > > Q10: *One more point related to theoretical novelty. In Theorem 6 of [arXiv:2006.11077] authors consider partial participation with any (fixed) distribution over nodes (section 4.2), where, in contrast to the current submission, communication is done in every iteration but in compressed form. Note that in that Theorem 6, (1) instead of uniformly bounded second moment assumption they use more relaxed one (assumption 3), (2) partial participation can potentially be biased and (3) more importantly, the rate is O(1/T) without any non-vanishing bias term. It seems that the non-vanishing bias term is eliminated by adjusting the aggregation step at the server. With this point, the presence of non-vanishing bias term in Theorem 3.1 is not justified by the usage of biased selection strategy as it seems to be a defect of analysis/method.*
> > > > >
> > > > > A10:
> > > > > We clarify the reviewer’s misunderstanding in points (2) and (3) above with the following points:
> > > > > - **[arXiv:2006.11077] considers an “effectively unbiased” client selection** because they normalize the gradient bias at the time of aggregation, and hence consider an unbiased gradient estimator. This is explicitly mentioned in the paper  [arXiv:2006.11077] as:
> > > > > "````The aggregation step … is adjusted to lead to an unbiased estimator of the gradient which gives, $\Delta_k=\sum_{i\in\mathcal{S}^k}\frac{1}{np_i}\Delta_i^k$."
> > > > > The normalization factor $p_i$ in the aggregation step above forces the aggregated gradient to be “unbiased” (which is not fixing or eliminating the non-vanishing bias, but simply considering a setting where there is no effective bias). The rate $\mathcal{O}(1/T)$ doesn’t have a non-vanishing term because the aggregated gradient is unbiased. Therefore their convergence analysis is actually analogous to that of an unbiased client selection with simple averaging. Our paper subsumes both unbiased gradient estimators, and because we consider biased gradient estimators to the picture, we introduce the convergence speed improvement along with the non-vanishing bias in our analysis.  Hence Theorem 3.1 is not an artifact of the analysis or the algorithm, but rather a property of biased client selection strategies. To summarize, [arXiv:2006.11077] and our paper take strictly different two settings.
> > > > > - Moreover, **[arXiv:2006.11077]’s scope of client selection strategies is limited than our paper’s scope.** The client selection probabilities presented in [arXiv:2006.11077] are fixed across communication rounds meaning the selection scheme is not aware of the global progress and non-time variant. Our paper includes client selection strategies with time-varying probabilities. For the aggregation scheme proposed in [arXiv:2006.11077] to work, the sampling probabilities in [arXiv:2006.11077] should be known apriori. For client selection strategies that are cognizant of the global model training progress, it is nearly impossible or very difficult to obtain the exact $p_i^{(t)}$ beforehand because it may depend on the loss values of clients and change over time, and the analysis in our paper doesn’t require to know these probabilities.

---

> > > > > > ### Comment · AnonReviewer4 · 2020-11-20
> > > > > > **Response to author(s)**
> > > > > >
> > > > > > Based on your definition of unbiased client selection, each client $k$ must be selected with probability $p_k=D_k/\sum_j D_j$ equal to the fraction of the data of client $k$. This definition is then used to show that selection skew metric $\rho$ is always $1$ for unbiased selection scheme. In [6], probability $p_k\in(0,1]$ of selecting client $k$ can be anything regardless of the local data size. Hence, based on your definition of unbiased client selection, this selection strategy cannot be unbiased for any values of $p_k$. Moreover, for the selection scheme of [6] the skewness metric $\rho$ depends on those probabilities $p_k$ and cannot be $1$ always. Hence, with aggregation step (2) and rate (7) of Theorem 3.1 we have non-vanishing term. However, with the adjusted aggregation $\Delta_k=\sum_{i\in S^k} \frac{1}{n p_i}\Delta_i^k$ and rate of Theorem 6 [6], there is no non-vanishing term.
> > > > > >
> > > > > > Unbiasedness of client selection and unbiasedness of gradient estimator are different. In [6] gradient estimator is unbiased, while selection scheme is allowed to be biased. In your paper, both are allowed to be biased. The message that there is a non-vanishing bias term in the rate (7) because of the biasedness of client selection does not seem to be true. To me, the real cause of the bias term in (7) is the biasedness of gradient estimator in (2) and [6] shows that (in some class of biased selection schemes) fixing the biasedness of gradient estimator eliminates the bias term.
> > > > > >
> > > > > > [6] Samuel Horváth, Peter Richtárik. A Better Alternative to Error Feedback for Communication-Efficient Distributed Learning, arXiv:2006.11077

---

> > > > > > > ### Author Response · Authors · 2020-11-20
> > > > > > > **Response to reviewer 4**
> > > > > > >
> > > > > > > Thanks for the response! It is true that the non-vanishing error term is due to the bias in the gradient estimate. Since we consider uniform aggregation of updates, the bias in the gradient estimate is indeed a direct result of biased client selection. However, preserving this bias is not a defect but rather a strength of our proposed client selection strategy. Selection bias (when added correctly) improves the convergence speed as shown by our theoretical analysis. However, it may add a non-vanishing bias term, which we observe as being small in our experiments. In the updated version of the paper, we propose the adapow-d strategy which gradually reduces the bias during training and achieves the best of both worlds -- it achieves faster convergence as well as a vanishing bias term. **[6] simply eliminates the bias in the gradient estimate, but as a result, also misses out on the potential convergence speed-up that biased gradients can provide.** Also, this method of eliminating bias only works for the case where $p_i$ is fixed across rounds and known a priori. Hence, it cannot be applied to our proposed strategy which adapts to the training progress.
> > > > > > >
> > > > > > > We hope that this explanation clarifies that our paper is distinct from [6]. Please let us know if you have any further questions.

---

### Official Review · AnonReviewer3 · 2020-10-27
**Client Selection in Federated Learning: Convergence Analysis and Power-of-Choice Selection Strategies**

**Rating:** 6
**Confidence:** 3

**Review:**

##########################################################################

Summary:

The paper analyzes a biased client selection policy in the context of the federated learning paradigm.
In particular, instead of random selection of the clients with a probability that is proportional to
the size of the local dataset, clients with higher local loss values are selected.
Despite being already used as heuristics, the authors claim that their work is the first one to
propose a convergence analysis of a biased client selection mechanism in the context of federated
learning. Under some (quite stringent) assumptions, the authors derive a global bound for the expected
error after $T$ iterations. The bound is composed of two terms: a vanishing term and a non-vanishing one
which comes as a consequence of the biased client selection mechanism. From Theorem 3.1 it emerges clearly
that the biased selection mechanism poses a trade-off: on one hand the non-vanishing term increases the more biased is the selection, on the other hand the vanishing one benefits from the biased selection mechanism since the more biased is the selection the faster it will decay to zero.
The biased selection mechanism comes with some additional computational and communication costs with respect to a fully random selection: the authors propose a couple of heuristics to alleviate these extra-costs.
In the experimental section the proposed method is evaluated on different benchmarks and with different datasets.

 ##########################################################################

Reasons for score:

The paper is well-written and the theoretical results look correct. At the same time, there are some weaknesses
(see the section on Cons).

##########################################################################

Pros:

1. The paper takes is well-written and really easy to follow.

2. The authors attempt to provide a more rigorous analysis to an already-in-use heuristic and it is the first time that such
analysis is carried out in this context.

3. The paper has a good balance across the sections and it touches on theory and experimental part in a balanced way.

##########################################################################

Cons:

 1. The assumptions on which the theoretical analysis is based are quite stringent and could be further relaxed (especially
strong convexity).

 2. Incoherent benchmarks: the authors are not reasoning regarding the introduced assumptions and whether they hold. It actually seems that at least in the DNN case the assumptions are not holding since the landscape is notoriously non-convex.

 3. Experiments only show the behavior of the heuristics pow-d. What happens when d=K? In order for the experiment to be in line with the theoretical results this case should be analyzed at least once. Introducing the extra random selection of $m\leq d\leq K$ clients could indeed change the setting and therefore the convergence.

 4. Way too strong and unjustified statements, i.e. faster convergence to a global minimum (according to Th.3.1. it does not converge), convergence up to 3 times faster (in one benchmark, might easily depend on hyperparameter settings or on the heuristic added on top on power-of-chance).

5. All the statements regarding the test accuracy are pure speculations as there are no theoretical and/or consistent and extensive empirical evidence which suggest that the biased selection mechanism leads to better generalization. Intuitively adding a bias in the selection should lead more easily to overfitting.


##########################################################################

Questions during rebuttal period:

Please address and clarify the cons above

#########################################################################

Interesting General Aspect Not Considered

1. local SGD iterations: how does $\tau$ impacts on the convergence? Is it beneficial to perform more local SGD steps or does it become harmful for the convergence? How does this relate with the biased selection mechanism?

2. first random selection of $d$ clients based on proportions of dataset: unbalanced work load might create inefficiency since the general method described is synchronized.

---

> ### Author Response · Authors · 2020-11-18
> **Response to reviewer 3 (reviewer's responses are in italics)**
>
> Thank you very much for the detailed review of the paper! We appreciate the reviewer's comment on our paper being balanced on both theory and experiments, well-written, and novelty of the analysis of biased client selection. We address the reviewer's concerns below. We hope the reviewer re-evaluates the score in favor of the paper!
>
> Q1: *The assumptions on which the theoretical analysis is based are quite stringent and could be further relaxed (especially strong convexity).*
>
> A1:
> - We acknowledge that our analysis is based on strongly-convex functions, and do not include non-convex ones. Our paper's significance is in the novelty of the first analysis for biased client selection strategies that have not been looked into before and the insight it gives to the FL community along with the effectiveness of our proposed power-of-d client strategy. To the best of our knowledge, there has been no work giving convergence insights for biased client selection strategies that are cognizant of the training process for either strongly-convex or non-convex functions. Our work presents the first insight in this area, and we intend to extend the analysis to non-convex scenarios as the next step. We would like to highlight that between two different sets of analysis techniques (strongly-convex and non-convex), considering the latter is outside the scope of the current paper.
>
> Q2: *Incoherent benchmarks: the authors are not reasoning regarding the introduced assumptions and whether they hold. It actually seems that at least in the DNN case the assumptions are not holding since the landscape is notoriously non-convex.*
>
> A2:
> - The reviewer is correct in that the DNN loss landscape is non-convex. While we do not claim that our convergence analysis holds in this case, the experiments on DNN show that the proposed client selection strategies do work well even for non-convex loss landscapes. The strongly-convex assumption does hold for the quadratic simulation results presented in Fig. 2. Note that we do not introduce any new assumptions that haven’t been used in previous literature. [1-4] also use the assumptions used in our paper to provide theoretical insight into distributed algorithms.
>
> - [1] Sebastian Urban Stich. Local sgd converges fast and communicates little. In ICLR 2019
> [2] Debraj Basu, Deepesh Data, Can Karakus, and Suhas Diggavi. Qsparse-local-SGD: Distributed SGD with Quantization, Sparsification, and Local Computations. In NeurIPS 2019
> [3] Xiang Li, Kaixuan Huang, Wenhao Yang, Shusen Wang, and Zhihua Zhang. On the convergence of fedavg on non-iid data. In International Conference on Learning Representations (ICLR), July 2020.
> [4] Yichen Ruan, Xiaoxi Zhang, Shu-Che Liang, and Carlee Joe-Wong. Towards flexible device participation in federated learning for non-iid data. ArXiv, 2020.
>
> Q3: *Experiments only show the behavior of the heuristics pow-d. What happens when d=K? In order for the experiment to be in line with the theoretical results this case should be analyzed at least once. Introducing the extra random selection of m≤d≤K clients could indeed change the setting and therefore the convergence.*
>
> A3:
> - We would like to clarify that our theoretical analysis is a general analysis that is applicable for any selection strategy $\pi$ that is cognizant of the training progress. It is not limited to the pow-d strategy or any specific value of $d$ such as $d=K$. The case when $m≤d≤K$ clients is subsumed in the convergence analysis. As $d$ increases, the $\overline{\rho}$ also increases along with the selection bias, as shown in Fig.2(b), being consistent with our theoretical results.
>
> - We show the effect of $d=K$ for the quadratic simulations in Fig.2 (a) and Fig.2(b) and the synthetic simulations for $K=30, m=3$ in Fig.3. We show that for $d=K$ which is the case for maximum selection bias, we achieve higher convergence speed but also the highest selection bias which is in line with our theoretical results.

---

> > ### Author Response · Authors · 2020-11-18
> > **(cont'd) Response to reviewer 3 (reviewer's responses are in italics)**
> >
> > Q4: *Way too strong and unjustified statements, i.e. faster convergence to a global minimum (according to Th.3.1. it does not converge), convergence up to 3 times faster (in one benchmark, might easily depend on hyperparameter settings or on the heuristic added on top on power-of-chance).*
> >
> > A4:
> > - We state that the convergence up to 3 $\times$ faster to a specific target global loss value, not the global minimum, as shown in the synthetic simulation result section. To clarify this, we modified figure 3 so that the figure better shows the faster convergence towards a certain global loss value. Moreover, for the DNN experiments, we show in Table 1 that pow-d can achieve the same test accuracy (=60%) approximately at least $2\times$ faster than the random sampling case. To justify that our results are not an artifact of hyperparameter running, we have added additional DNN experiments for different mini-batch sizes and local epoch values in the appendix of the updated paper. Moreover, we have added an additional benchmark experiment on the larger dataset EMNIST ('by digits') to further strengthen the validation of the performance of pow-d.
> >
> > Q5: *All the statements regarding the test accuracy are pure speculations as there is no theoretical and/or consistent and extensive empirical evidence which suggests that the biased selection mechanism leads to better generalization. Intuitively adding a bias in the selection should lead more easily to overfitting.*
> >
> > A5:
> > - Note that the proposed client selection strategies aim to speed-up training and our theoretical analysis also aims to show an improvement in the convergence of training loss. We do not claim a provable improvement in test accuracy. The test accuracy experiments are presented just as a sanity check to show that the proposed biased client selection strategies do not lead to overfitting. Although our proposed pow-d strategy adds bias, this bias is expected to reduce during the course of training as the local loss values are equalized across clients. Intuitively, this can prevent overfitting, leading to better generalization as the test accuracy results suggest. To strengthen the consistency of empirical evidence, as mentioned above, we added additional experiments on different hyperparameters (mini-batch size and local epochs), and with a different dataset EMNIST.
> >
> > Suggestion 1: *local SGD iterations: how does τ impacts on the convergence? Is it beneficial to perform more local SGD steps or does it become harmful for the convergence? How does this relate with the biased selection mechanism?*
> >
> > Response to suggestion 1:
> > - We thank the reviewer for the suggestion! We evaluated the effect of $\tau$ for different level of data heterogeneity in our DNN experiments with FMNIST and added the results to our updated paper. Results show that for lower data heterogeneity ($\alpha=2$), increasing $\tau$ is beneficial for pow-d because the statistical discrepancies of datasets across clients is small, and biasing actually helps the generalization performance. Therefore performing more local updates to the selected clients with higher losses allows pow-d to reach convergence faster without the negative effect of selection bias. However, in the case of larger data heterogeneity ($\alpha=0.3$), larger $\tau$ increases the negative effect of selection bias due to the increased statistical discrepancies of datasets across clients, and allows the baseline random selection strategy to perform similarly with pow-d.
> >
> > Suggestion 2: *first random selection of d clients based on proportions of dataset: unbalanced workload might create inefficiency since the general method described is synchronized.*
> >
> > Response to suggestion 2:
> > - Thank you for the interesting suggestion. We can illustrate the problem of unbalanced workload across the selected $d$ clients and overcome stragglers by doing the following: select the $d$ clients as in the pow-d scheme, and then allow $L (d<L<m)$ clients to perform local updates and the server only receives the $m$ updates that finish first to update the global model. Here, $L$ should preferably not be too large (ex: $L\approx1.2m).

---

### Official Review · AnonReviewer1 · 2020-10-28
**Borderline paper; Needs more experiments**

**Rating:** 6
**Confidence:** 4

**Review:**

##########################################################################

Summary:

This paper studies federated learning (FL) and proposes nonuniform sampling of participating clients. Clients are selected according to their losses; clients with big losses are likely selected. The proposed algorithm has a rigorous convergence analysis. The proposed algorithm is demonstrated powerful on small scale datasets.



##########################################################################

Reasons for score:

I find this submission a borderline paper. I am fine with either acceptance or rejection. This paper has theoretical guarantees and strong empirical advantages. But the actually used algorithm may not be very useful. The experiments are conducted on small datasets. I am not confident whether this work will have an advantage on large-scale data.




##########################################################################

Pros:

+ The idea of choosing participating clients according to loss is an interesting and reasonable idea.

+ The non-uniform sampling of clients is proved to converge.

+ The nonuniform sampling algorithm has a strong empirical advantage.


##########################################################################

Cons:

1. The actually used algorithm is the power-of-choice strategy. It firstly uses a uniformly sampled set of $d$ clients to compute losses and then sample a subset of $m < d$ clients out of the $d$ clients.
Here is my question: The server needs to communicate with the $d$ clients from the candidate set. The server must wait for their responses. So the $d$ clients are actually active. Why not use all the $d$ clients to compute gradients? Is the proposed strategy comparable to using all the $d$ clients?

2 FMNIST seems to be the only used dataset. The dataset is not big enough. Even if it is big enough, I would like to see empirical results on other datasets, e.g., MNIST, CIFAR10, mini-ImageNet, etc. Admittedly, this work has an advantage on FMNIST. But I am not convinced that this work is better in general.



##########################################################################

---

> ### Author Response · Authors · 2020-11-18
> **Response to reviewer 1 (reviewer's responses are in italics)**
>
> We greatly thank the reviewer for the positive feedback on our paper. The reviewer raised two issues: 1) why the pow-d strategy does not use all d clients for training, and 2) that the experiments on the FMNIST dataset are not enough. We address both these concerns below.
>
> Q1: *The actually used algorithm is the power-of-choice strategy. It firstly uses a uniformly sampled set of d clients to compute losses and then sample a subset of m<d clients out of the d clients. Here is my question: The server needs to communicate with the d clients from the candidate set. The server must wait for their responses. So the d clients are actually active. Why not use all the d clients to compute gradients? Is the proposed strategy comparable to using all the d clients?*
>
> A1:
> - First, having the $d$ clients actually perform the local updates rather than just send their local loss values will incur a significantly higher computation cost than performing updates at only $m < d$ clients. We highlight that pow-d does not require $d$ clients to actually perform gradient updates, but just requires inference tasks on the training data for the loss values.
> - Moreover, to address the possible communication and computation cost for the loss inference task of the $d$ clients, in the paper we propose rpow-d, which simply uses the last received loss values from the previously selected clients to select $m$ clients from the selected $d$ clients in the subset. rpow-d requires no additional computation or communication cost compared to the baseline random strategy, but performs better than the baseline as shown in our DNN experiments.
> - Lastly, we emphasize that pow-d does not uniformly sample the final $m$ clients, but sample them according to their importance (i.e., loss values). Therefore, our strategy is distinguished from simply uniformly sampling in proportion to dataset size of $d$ clients and performing local updates.
>
> Q2: *FMNIST seems to be the only used dataset. The dataset is not big enough. Even if it is big enough, I would like to see empirical results on other datasets, e.g., MNIST, CIFAR10, mini-ImageNet, etc. Admittedly, this work has an advantage on FMNIST. But I am not convinced that this work is better in general.*
>
> A2:
> - For more experimental validation, since the MNIST and CIFAR10 datasets have the same size as the FMNIST dataset considered in our experiments, we have added experiments on a larger dataset EMNIST ("by Digits"). The EMNIST dataset is 4 times larger (240,000 training data with 40,000 testing data)  than FMNIST, MNIST, and CIFAR10 (which have 60000 training samples each).
> - We show consistency in the effective performance in training loss and test accuracy results for our proposed pow-d strategy. Note that the performance improvement offered by the pow-d strategy and its variants is due to data heterogeneity across clients and not specific to one particular dataset. So we expect to see similar performance on other datasets. Moreover, we stress that the experiments are a sanity check for the performance of our proposed pow-d strategy and its variants. Our paper’s main novelty and contribution is that it gives the first convergence analysis of biased client selection strategies, which has not been investigated before.

---

> > ### Comment · AnonReviewer1 · 2020-11-24
> > **Reply**
> >
> > (1) The authors said in A1: "First, having the d clients actually perform the local updates rather than just send their local loss values will incur a significantly higher computation cost than performing updates at only $m<d$ clients."
> >
> > The computations are performed in parallel. More computation does not make things worse.
> >
> > (2) Q1 & A1: The authors did not directly answer my question. I am interested in the comparisons of "non-uniformly sampling $d$ clients" (this work) and "uniformly sampling $m$ clients" (standard). What are their strength and weakness? What are the costs of their communication complexities and latencies? What are their empirical performances?
> >
> > In practice, a big portion of the clients are inactive, which means high latency and long response time. Sampling a small subset of clients can alleviate the problem. But the proposed method does not alleviate the problem, as it communicates with $d$ clients.
> >
> > (3) I rarely see optimization papers with experiments on only one or two datasets, unless the studied algorithms are standard. I insist that more datasets must be used. EMNIST alone is not enough because digit recognition is an over-easy problem. Even if you use the MNIST 8 Million Dataset, the results do not imply that in general, this work is useful. Natural image datasets must be used. Natural language datasets will be a plus.

---

> > > ### Author Response · Authors · 2020-11-24
> > > **Response to reviewer 1 (reviewer's responses are in italics)**
> > >
> > > *The authors said in A1: "First, having the d clients actually perform the local updates rather than just send their local loss values will incur a significantly higher computation cost than performing updates at only $m<d$ clients."
> > > The computations are performed in parallel. More computation does not make things worse.*
> > >
> > > AA1:
> > > - Although the computations are performed in parallel, the total computation cost of all $d$ clients will still be linearly increasing in $d$. However, even if we go by the reviewer’s argument that this does not count as an additional cost, another problem with using more clients is that we can suffer from straggling delays stemming from waiting for the slowest client to finish their local computation -- local computation times can indeed vary across clients. As $K$ gets larger, which is a realistic scenario of FL, the problems of stragglers can severely deteriorate the communication efficiency if we perform local updates on all $d$ clients.
> > >
> > > *Q1 & A1: The authors did not directly answer my question. I am interested in the comparisons of "non-uniformly sampling $d$ clients" (this work) and "uniformly sampling $m$ clients" (standard). What are their strength and weakness? What are the costs of their communication complexities and latencies? What are their empirical performances?
> > > In practice, a big portion of the clients are inactive, which means high latency and long response time. Sampling a small subset of clients can alleviate the problem. But the proposed method does not alleviate the problem, as it communicates with $d$ clients.*
> > >
> > > AA2:
> > > - In Table 1 in our original paper, we present results on “non-uniformly sampled $d$ clients" and “uniformly sampled $m$ clients” where d=6<m=10 and d=6>m=3. For d<m, we show that in terms of communication cost, our work actually takes less communication rounds (=89)  to reach the test accuracy 60\% than the latter (standard, =172). Moreover we show the empirical test accuracy performance is slightly higher for our work than the standard uniform sampling. Therefore, even if d<m, selecting $d$ clients to perform pow-d can be more communicationally efficient than performing local updates on $m$ with uniform sampling. For d>m, we show that this performance improvement regarding communication rounds is even more improved than the case of d<m. In addition, note that our proposed variant of pow-d, rpow-d, doesn’t require any additional communication cost compared to the standard uniform sampling, but still is able to perform better than the baseline.
> > >
> > > *I rarely see optimization papers with experiments on only one or two datasets, unless the studied algorithms are standard. I insist that more datasets must be used. EMNIST alone is not enough because digit recognition is an over-easy problem. Even if you use the MNIST 8 Million Dataset, the results do not imply that in general, this work is useful. Natural image datasets must be used. Natural language datasets will be a plus.*
> > >
> > > AA3:
> > > - In your initial review, you suggested using MNIST, CIFAR10, or mini-ImageNet etc, for a larger dataset size and did not specify which one you would like to be added, which is why we went with EMNIST (‘by digits’) that had the largest dataset size amongst the three. We agree with the reviewer that experiments on natural image/language datasets will be a great addition to the paper. We already had FMNIST experiments in the original version. We can add CIFAR10 experiments and Shakespeare dataset experiments in the final version.
> > >
> > > - However, since the reviewer did not ask for these experiments earlier (we uploaded our initial response on Nov 18th), we are not able to run additional experiments before the end of the author response period. We request that the reviewer not use this as the reason to reject the paper. We stress again that the paper’s main contribution is the novel insights on the effect of biased client selection on the convergence of federated learning, and the proposal of a new client selection strategy. The ML experiments are primarily done as a sanity check of our proposed scheme to ensure consistency with the theory.

---

> > > > ### Comment · AnonReviewer1 · 2020-11-24
> > > > **Reply**
> > > >
> > > > If the authors promise to add to the final version extra experiments on natural images (e.g., CIFAR) and natural language, I won't recommend rejection.

---

> > > > > ### Author Response · Authors · 2020-11-24
> > > > > **Thanks for the reply!**
> > > > >
> > > > > We thank the reviewer for the helpful comments, and we will run and add the additional experiments for our final version on each natural image and natural language dataset.

---

### Official Review · AnonReviewer2 · 2020-11-03
**Review of Client Selection in Federated Learning: Convergence Analysis and Power-of-Choice Selection Strategies**

**Rating:** 6
**Confidence:** 4

**Review:**

This paper analyzes the convergence of FedAvg with biased client selection strategies. And the new strategy power-of-choice is designed based on the convergence results. This new strategy is numerically compared with two benchmark strategies and shows faster convergence and higher testing accuracy.

Pros:
1. The convergence analysis is new and novel. This paper provides the first convergence analysis for a general class of biased client selection strategies. New concept, selection skew, is introduced as the measure of strategies. Based on the new concept, the analysis quantifies how the bias of the strategy affects the convergence speed of FedAvg.
2. New strategy pow-d is proposed based on the insights of the convergence analysis. The performance of pow-d is impressive in the FMNIST experiment, where it significantly improves the test accuracy under random selection strategy.

Cons:
1. Although the analysis is interesting and insightful, the effect of \rho seems to be limited. There are three constant terms in the vanishing error term in Eq.(7). And only the first term will be decreased when increasing \bar{\rho}. However, it is not obvious if the first term is the dominating term.
2. Is it possible to give an exact or estimated values of \bar\rho and \tilde\rho for different strategies in the numerical experiments?

---

> ### Author Response · Authors · 2020-11-18
> **Response to reviewer 2 (reviewer's responses are in italics)**
>
> Thank you very much for highlighting the paper’s key novel contributions: 1) that we provide the first convergence analysis of biased client selection strategies and 2) the proposed pow-d strategies gives an impressive improvement in the FMNIST experiments. Our responses to both the concerns are given below.
>
> Q1: *Although the analysis is interesting and insightful, the effect of \rho seems to be limited. There are three constant terms in the vanishing error term in Eq.(7). And only the first term will be decreased when increasing \bar{\rho}. However, it is not obvious if the first term is the dominating term.*
>
> A1:
> - We show that the first term $\frac{4L (32\tau^2 G^2+\sigma^2/m)}{ 3\mu^2\overline{\rho}}$ is indeed significant in the vanishing error term,  below, $\frac{1}{(T+\gamma)}\left[\frac{4L (32\tau^2 G^2+\sigma^2/m)}{ 3\mu^2\overline{\rho}}+\frac{8L^2\Gamma}{\mu^2}+\frac{L\gamma\|\overline{\mathbf{w}}^{(0)}-\mathbf{w}^*\|^2}{2}\right]$, through experiments held out in three different settings: quadratic (strongly-convex case), synthetic dataset (logistic regression problem), and DNN with FMNIST dataset. Note that the quadratic simulation directly reflects the theory due to the quadratic function's strongly-convex property.  Especially in the quadratic and synthetic simulations we show that biasing more towards the clients with larger loss, i.e., larger $\overline{\rho}>1$, leads to faster convergence. If the first term not was not dominant, larger $\overline{\rho}>1$ would not have led to faster convergence due to the other two terms in the vanishing error term. Figure 2 shows how $\overline{\rho}$ actually matters with estimated $\overline{\rho}$ values by showing the convergence speed up.
>
> - In addition to the point above, note that the first term is the only term that can be controlled through client selection (the number of local updates $\tau$, stochastic gradient bound $G$, and variance bound $\sigma$). The other two terms are properties of the objective function, dataset distributions, and the initial point which cannot be controlled. Therefore we believe that the decrease by a factor of $\overline{\rho}$ in the first term is sufficient to show the positive effect of biasing client selection.
>
> Q2: *Is it possible to give an exact or estimated values of \bar\rho and \tilde\rho for different strategies in the numerical experiments?*
>
> A2:
> - This question is already answered in Fig 2(b), where we plot the estimated values of $\overline{\rho}$ and $\widetilde{\rho}/\overline{\rho}$ for varying $d$ and $K$ and for different client selection strategies. Perhaps the reviewer missed seeing it -- we will make it more prominent in an updated version. Our theory matches well with the quadratic simulations in that as $d$ increases the estimated $\overline{\rho}$ increases but the selection bias $\widetilde{\rho}/\overline{\rho}$ also increases. The methodology of getting these estimated theoretical values is elaborated in Appendix F of the original paper.

---

### Author Response · Authors · 2020-11-18
**General response to all reviewers**

We thank all the reviewers for the valuable feedback! We updated our paper reflecting the reviewers' constructive feedback, and made the changes be seen by **magenta** color. All the reviewers highlight the novelty of our paper -- that it is the first convergence analysis of biased client selection strategies in federated learning, and that the proposed power-of-choice strategies give an impressive speed-up in training time. The reviewers did not point out any major issue with the paper, but suggested some clarifications and improvements in the algorithm and the experiments. We have responded to each reviewer’s feedback in the individual comments below, and will upload the fully updated version of the paper soon accordingly. The main changes we made are:

1. Added DNN experiment on larger dataset EMNIST (‘by digits’) with larger number of clients $K=500,~C=0.03$ to evaluate pow-d’s performance compared to the baseline strategy and show consistency in performance improvement with the original FMNIST experiment.
2. Performed additional experiments with different number of local updates and mini-batch sizes to evaluate effect of hyperparameters.
3. Added the proposition and evaluation of ‘adapow-d’ which is an extension of pow-d that modulates $d$ during training so that selection bias can be eliminated in the end and converge to the global minimum while still enjoying the fast convergence property in the initial phase of training with $d>m$.

Please let us know if there are any additional questions that we can address during the remaining discussion period. We hope that the reviewers and the area chair will judge the paper favorably after reading our responses.

---

### Decision · Program_Chairs · 2021-01-07
**Final Decision**

**Decision:**

Reject

**Comment:**

In federated learning, distributed and resource-limited client nodes cooperatively train a model without sharing their local data. The results thus far on analyzing the  convergence of federated learning are restricted to “unbiased” client participation, where the probability of a client c being selected is proportional to c’s data size. This work presents the first convergence analysis of federated learning for biased client selection, and quantifies the impact of selection skew on time to convergence. Specifically, biasing toward clients with higher local loss is shown to be beneficial, and a protocol is developed based on this, to trade between convergence time and solution bias.

The paper is in general well-written, and develops a natural idea.

The strong-convexity assumption is a concern: how much can it be weakened? The authors are also asked to run experiments systematically on (much) larger datasets. The test-accuracy and possible-overfitting concerns also need to be addressed in more depth. The authors are also encouraged to see how much Assumption 3.4---uniformly-bounded stochastic gradients---can be dispensed with.